



# Warm-phase Microphysical Evolution in Large Eddy Simulations of Tropical Cumulus Congestus: Constraining Drop Size Distribution Evolution using Polarimetery Retrievals and a Thermal-Based Framework

McKenna W. Stanford[1,2,†], Ann M. Fridlind[2], Andrew S. Ackerman[2], Bastiaan van Diedenhoven[3], Qian Xiao[4], Jian Wang[4], Toshihisa Matsui[5,6], Daniel Hernandez-Deckers[7], and Paul Lawson[8]

[1]Center for Climate Systems Research, Columbia University, New York, NY, USA
[2]NASA Goddard Institute for Space Studies, New York, NY, USA
[3]SRON Netherlands Institute for Space Research, Utrecht, the Netherlands
[4]Center for Aerosol Science and Engineering, Department of Energy, Environmental and Chemical Engineering, Washington University in St. Louis, Saint Louis, MO, USA
[5]Mesoscale Atmospheric Processes Laboratory, NASA Goddard Space Flight Center, Greenbelt, MD, USA
[6]Earth System Science Interdisciplinary Center – ESSIC, University of Maryland, College Park, MD, USA
[7]Grupo de Investigación en Ciencias Atmosféricas, Departamento de Geociencias, Universidad Nacional de Colombia, Bogotá, Colombia
[8]Stratton Park Engineering Company, Inc., Boulder, CO, USA
[†]Current affiliation: Atmospheric, Climate, and Earth Sciences Division, Pacific Northwest National Laboratory, Richland, WA, USA

**Correspondence:** McKenna W. Stanford (mckenna.stanford@pnnl.gov)

**Abstract.**

Improving parameterizations of convective microphysics in Earth system models (ESMs) requires well-constrained cases suitable for scaling between cloud-resolving models and ESMs. We propose a benchmark large eddy simulation (LES) cumulus congestus case study from the NASA Cloud, Aerosol, and Monsoon Processes Philippines Experiment (CAMP²Ex)

and demonstrate its observational constraints using novel polarimetric retrievals and in situ cloud microphysics measurements. Simulations using bulk and bin microphysics with observed aerosol input are compared to cloud-top retrievals of cloud droplet effective radius ($R_{\text{eff}}$) and number concentration ($N_d$) from the airborne Research Scanning Polarimeter (RSP). The bulk scheme reasonably reproduces characteristic profiles of cloud-top $N_d$ that decrease with altitude, while the bin simulation realizes greater discrepancies due to weaker precipitation formation. The $N_d$ profile is strongly sensitive to the collision-coalescence

process and the vertically resolved aerosol distribution, but appears well-constrained, whereas a persistent low-bias in $R_{\text{eff}}$ is evident in both schemes. Comparison of simulated and in situ droplet size distributions (DSDs) show that low-biased $R_{\text{eff}}$ originates from a cloud droplet mode that is too narrow relative to observations. Finally, a thermal-tracking framework demonstrates that the dilution of $N_d$ throughout a thermal's lifetime is heavily determined by collision-coalescence and the height-varying aerosol distribution, and that in the absence of these, the impact of entrainment on diluting $N_d$ is largely offset by continuous

aerosol activation. Implications for developing warm-phase convective microphysics schemes for ESMs, evaluation of cumulus



congestus using single column model versions of ESMs, and translating results to global, space-based polarimetry platforms are discussed.

## 1 Introduction

Cumulus congestus clouds play an important role in the global water and energy budget. In the tropics, they represent the
intermediate mode of the trimodal tropical convection distribution in between shallow trade-wind cumuli and deep convection
(Johnson et al., 1999). Definitions vary in the literature, but cumulus congestus generally have cloud top heights (CTHs)
between 4 and 8 km that are either stabilized around the 0 °C level (so-called "terminal congestus") or penetrate the 0 °C
level with sustained vertical growth ("transient congestus"). Wall et al. (2013) showed using 5 years of CloudSat profiles that
congestus contribute up to 12% of the total cloud population in the tropics and up to 18% of all clouds with tops lower than 8 km
over regions such as the Amazon, central Africa, and the maritime continent. During the Tropical Ocean Global Atmosphere
Coupled Ocean Atmosphere Response Experiment (TOGA-COARE) field campaign (Webster and Lukas, 1992) conducted
over the western Pacific warm pool, Johnson et al. (1999) concluded that 57% of precipitating convective clouds were identified
as congestus and contributed to 28% of total convective rainfall. Transient congestus, which were shown to account for ∼ 30-
40% of congestus clouds observed by CloudSat by Luo et al. (2009), are also important for promoting growth to deeper
convective clouds (Kuang and Bretherton, 2006; Waite and Khouider, 2010; Hohenegger and Stevens, 2013). Understanding
their dynamical and microphysical evolution is therefore crucial for developing cumulus and convection parameterizations
for large-scale models that account for their contributions to global precipitation and role in redistributing heat, momentum,
aerosol, and moisture throughout the troposphere.

Mechanistically, congestus dynamics and microphysics are intricately linked. Cumulus clouds are composed of numerous
thermals with relatively short lifetimes (3-5 minutes; Hernandez-Deckers and Sherwood, 2016, 2018; Matsui et al., 2024) that
successively rise to the thermodynamic neutral buoyancy level, unless their ascent is precluded by the effects of dry-air en-
trainment. Supersaturation in these thermals acts as the primary source of condensation. However, thermals generate toroidal
circulations that enhance cloud dilution on the inflow branch via entrainment of relatively dry environmental air, therefore
additionally acting to evaporate condensate (e.g., Lasher-Trapp et al., 2005; Moser and Lasher-Trapp, 2017; Morrison et al.,
2020; Peters et al., 2020; Pardo et al., 2020). Chandrakar et al. (2021) used a detailed Lagrangian microphysics model to sim-
ulate relatively shallow cumulus (CTHs < 5 km) and found that entrainment of aerosols by thermal circulations also played a
significant role in secondary activation (activation of cloud droplets above cloud base)–another condensate source. Entrainment
and secondary activation are two mechanisms that contribute to the broadening of drop size distributions (DSDs) with altitude
as a thermal rises, in addition to the collision-coalescence process and condensational processes. DSD broadening with height
has been evaluated extensively in large eddy simulations (LES) of cumulus (e.g., Lasher-Trapp et al., 2005; Cooper, 1989;
Grabowski and Abade, 2017; Morrison et al., 2018; Chandrakar et al., 2021), is supported by theory (Cooper, 1989), and has
been documented observationally in cumulus clouds (Warner, 1969; Manton, 1979; Lawson et al., 2015, 2017, 2022). Broad-
ening mechanisms additionally impact ice production through ice multiplication processes. Laboratory studies of secondary





ice production by a drop-shattering process indicate more numerous tiny splinters emitted during multiplication and more

frequent fragmentation with increased size of the frozen drop (e.g., Phillips et al., 2018; Lauber et al., 2018). Because DSD broadening is generally coincident with the production of larger drop sizes that reach the drizzle size regime, the initiation of ice multiplication once the drops are lofted may be dependent on the efficiency of the broadening mechanism. Indeed, Lawson et al. (2015, 2017, 2022) showed observational evidence of fractured frozen drops and spicules (indicative of a drop-shattering event–see Keinert et al. (2020)) in aircraft measurements of tropical cumulus congestus where DSD broadening with height

was also observed. On the other hand, copious ice production was not observed in a high-based congestus case sampled over the United Arab Emirates (UAE) shown in Morrison et al. (2022) where DSDs never showed substantial precipitation-sized drops.

In this work we evaluate the microphysical evolution of a tropical cumulus congestus case study using LES and robust observational constraints. Here we focus only on the liquid phase and the DSD evolution using a thermal-based framework.

Future work will build on this foundation, extending evaluation to the ice phase and exploring the role of ice multiplication. The selected case was observed during the NASA Cloud, Aerosol, and Monsoon Processes Philippines Experiment (CAMP²Ex) aircraft-based field campaign (Reid et al., 2023) and is representative of a field of cumulus congestus with growing tops that eventually realize an organized structure with cloud tops reaching $\sim$ -15 °C. Simulations are performed using both bulk and bin microphysics schemes and are constrained in a number of ways. First, observed aerosol particle size distributions (PSDs)

are used as input to represent trimodal lognormal distributions that vary with height. Second, large-scale thermodynamic and vertical motion conditions are harvested from a nested mesoscale simulation. Third, cloud-top $N_d$ and drop effective radius ($R_{\mathrm{eff}}$) are constrained using retrievals from the airborne Research Scanning Polarimeter (RSP; Cairns et al., 1999) and in-situ microphysics measurements. The multi-angle, multi-wavelength RSP measures total and polarized reflectance at cloud top and allows retrieval of $R_{\mathrm{eff}}$ and effective variance ($\nu_{\mathrm{eff}}$) using the sharply-defined cloud-bow at scattering angles in the

rainbow region of the visible spectrum. The retrieval of both $R_{\mathrm{eff}}$ and $\nu_{\mathrm{eff}}$ gives sufficient information about the DSD to retrieve $N_d$ with relatively few assumptions (Sinclair et al., 2019). Importantly, more RSP retrievals were available during the CAMP²Ex campaign than in any prior campaign. To this end, we evaluate the utility of spatiotemporally expansive RSP retrievals to supplement in situ aircraft transects in representing the warm-phase microphysical evolution of congestus, which to our knowledge is the first such study to do so. Finally, we incorporate a thermal-tracking framework at high temporal

frequency (Hernandez-Deckers and Sherwood, 2016, 2018) to isolate microphysical processes occurring within thermals and their contribution and control over the evolving DSD.

Cloud-top $N_d$ and $R_{\mathrm{eff}}$ retrievals and in situ DSD measurements during this CAMP²Ex case study indicate agreement with past studies in which DSDs broaden with altitude, $N_d$ decreases with height, and $R_{\mathrm{eff}}$ increases with height. Sensitivity tests are used to determine the processes that control such distinctive profiles. Here, we consider three primary controls on cloud-top

$N_d$/$R_{\mathrm{eff}}$ profiles: (1) the height-resolved aerosol PSD (which implicitly includes nucleation and condensational growth), (2) the efficiency of collision-coalescence and its parameterization in different warm-rain formulations, and (3) entrainment and mixing. Morrison et al. (2022) evaluated bin microphysics LES of a high-based congestus case from the UAE and compared it to in situ aircraft observations. They explored the role of collision-coalescence, secondary activation, aerosol loading, giant





cloud condensation nuclei (CCN), and entrainment on DSD evolution. Notably, the case they studied lacked development of large precipitation-sized drops, finding that warm-rain generation was more controlled by the sub-cloud aerosol distributions than by secondary activation and that evidence of dilution as essential for warm-rain formation was limited, though this was likely case-specific. Here, we investigate a case that produced substantial precipitation-sized drops and the mechanisms leading to that process.

An additional key component of this work is to develop a well-constrained LES case study for use as a benchmark in evaluating physics schemes in single-column model (SCM) versions of large-scale models, for example as has been demonstrated for subtropical marine stratocumulus-to-cumulus transitions (Sandu and Stevens, 2011; Neggers, 2015; Neggers et al., 2017), Arctic mixed-phase boundary layer clouds (Klein et al., 2009; Fridlind and Ackerman, 2017), and the currently ongoing cold air outbreak LES-SCM intercomparison project (Juliano et al., 2022). While LES intercomparison projects have been successfully carried out for case studies of shallow trade cumulus (Siebesma et al., 2003), precipitating trade cumulus (VanZanten et al., 2011), continental cumulus (Vogelmann et al., 2015; Endo et al., 2015; Lin et al., 2015), and tropical deep convection (Fridlind et al., 2012; Varble et al., 2011), considerably less attention has been paid to the congestus regime. Therefore, we seek to propose the presented case study as a plausible option for future congestus intercomparison projects with observational constraints provided by environmental aerosol measurements, thermodynamic and large-scale vertical motion derived from mesoscale simulations, and novel polarimetric retrievals that supplement in situ measurements of warm-cloud microphysics. Furthermore, using the thermal-based framework, we investigate implications for developing convective microphysics in large-scale models that consider entrainment from a viewpoint that diverges from standard entraining plume models.

The remainder of the article is structured as follows. Observations including environmental aerosol and in situ cloud measurements, RSP retrievals, and a description of the case study are provided in Section 2. The LES setup and experimental design are given in Section 3 and results are presented in Section 4. Finally, a discussion of using this case as a benchmark for tropical congestus SCM studies and additional implications for convective microphysics parameterization development and translation to space-based polarimetry platforms are provided in Section 5, followed by conclusions in Section 6.

## 2 Observations

Observations were obtained during the National Aeronautics and Space Administration (NASA) Cloud, Aerosol, and Monsoon Processes Philippines Experiment (CAMP[2]Ex; Reid et al., 2023). CAMP[2]Ex was an aircraft-based field campaign held from 25 August - 5 October 2019 and based out of the Clark International Airport, Philippines. Aircraft included NASA's P-3B and the Stratton Park Engineering Company's (SPEC) Learjet 35, which carried out 17 and 13 science flights, respectively. The goal of CAMP[2]Ex was to characterize the interaction of clouds, aerosol, and radiation in the monsoon system of Southeast Asia's maritime continent. Flights sampled a range of cloud conditions from shallow cumulus to deep convective systems. Here, we focus on a cumulus congestus event that occurred on 25 September 2019.



## 2.1 Aerosols: Fast Integrated Mobility Spectrometer (FIMS)

Aerosol size distributions with mobility diameter ranging from 10 to 600 nm were measured at 1 Hz resolution by the Fast Integrated Mobility Spectrometer (FIMS; Wang et al., 2017b, a, 2018; Kulkarni and Wang, 2006) onboard the NASA P-3B aircraft. FIMS operates by simultaneously detecting particles of different sizes based on the displacement of charged particles in an electric field. Calibration was performed before and after the CAMP$^2$Ex campaign. Calibrations of sizing accuracy and detection efficiency followed the procedure described in Wang et al. (2017a). Aerosol PSDs ranging from 10 to 600 nm were derived in 30 size bins with a bin width ($\Delta \log_{10} D_p$) equal to 0.061, where $D_\mathrm{p}$ is the single particle diameter, using the inversion technique of Wang et al. (2018).

Aerosol representation in numerical simulations follows from the methodology of Fridlind et al. (2017). Out-of-cloud FIMS measurements were composited over altitude ranges (Fig. 1b-c) and fitted to a distribution by minimizing the sum of squared residuals as follows:

$$\text{Residual} = \sum (F_i - \text{Obs}_i)^2 \tag{1}$$

where $F_\mathrm{i}$ is the fitted concentration and $\text{Obs}_\mathrm{i}$ is the FIMS concentration for size bin $i$. Three lognormal modes are then derived with a lognormal distribution represented by:

$$dN/d\ln D_p = \frac{N}{\sqrt{2\pi} \ln(\sigma_g)} \exp\left[-\frac{\ln^2(D_p/D_g)}{2 \ln^2(\sigma_g)}\right] \tag{2}$$

where $D_\mathrm{g}$ is the geometric mean particle diameter and $\sigma_g$ is the geometric standard deviation. Aerosol number concentrations ($N_\mathrm{a}$) vary with height at 0.5-km increments (Fig. 1a) while $D_\mathrm{g}$ and $\sigma_g$ are held constant with height. The realized trimodal $D_\mathrm{g}$ are 27, 62, and 153 nm, which roughly correspond to nucleation, Aitken, and accumulation modes, respectively. A constant hygrosopicity parameter ($\kappa$) is assumed for all three modes. The value is derived from time-averaged aerosol mass spectrometer (AMS) measurements for this research flight. We assume that aerosols consist of only $(NH_4)_2SO_4$ as the inorganic component with $\kappa = 0.53$ and an organic species with $\kappa = 0.1$ (Petters and Kreidenweis, 2007). The mass fractions of both components are converted to volume fractions using densities of 1.77 g cm$^{-3}$ for $(NH_4)_2SO_4$ and 1.4 g cm$^{-3}$ for average organic components (Hallquist et al., 2009). Consequently, the AMS-based hygroscopicity parameter using Equation (7) of Petters and Kreidenweis (2007) is 0.4.

## 2.2 Clouds: Research Scanning Polarimeter (RSP) retrievals

Retrievals of cloud-top $N_d$ and $R_{\text{eff}}$ are performed using the airborne Research Scanning Polarimeter (RSP; Cairns et al., 1999). The RSP makes total intensity and polarimetric measurements in nine spectral bands in the visible/near infrared and short-wave infrared. Onboard the P-3B aircraft, the RSP scans a given point on a cloud from multiple viewing angles (Alexandrov et al., 2012a). The CTH is retrieved using a multi-angle parallax method (Sinclair et al., 2017). Single-scattered light between scattering angles of 135° and 165° describe the sharply-defined cloud-bow in polarized reflectance that is used to retrieve the cloud-top $R_{\text{eff}}$ and $\nu_{\text{eff}}$ of the DSD (Alexandrov et al., 2012a). Cloud optical depth is retrieved from near-nadir reflectance measurements at a wavelength of 865 nm. Further details of RSP retrievals are provided by, e.g., Sinclair et al. (2017, 2020, 2021).

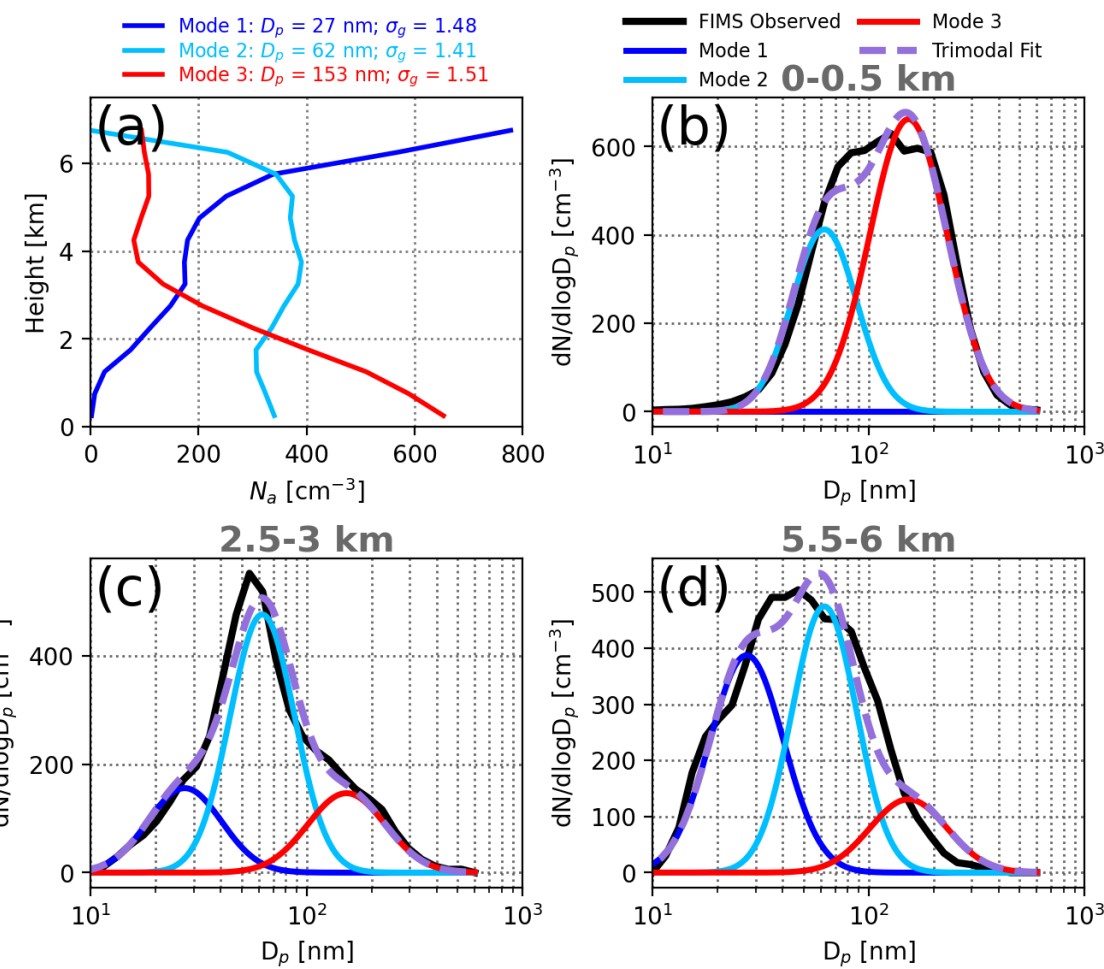

**Figure 1.** (a) Profiles of aerosol number concentration ($N_a$) for three lognormal modes with height-invariant geometric mean particle diameter ($D_p$) and geometric standard deviation ($\sigma_g$) listed above the panel. (b)-(d) Lognormal aerosol size distributions measured by the fast integrated mobility spectrometer (FIMS, black), the three derived modes (blue, light blue, and red), and the sum of all modes (dashed purple) for three example altitude ranges that span sub-cloud to upper entrainment environments (0-0.5 km, 2.5-3 km, and 5.5-6 km).



Droplet number concentrations are derived from the $R_{\text{eff}}$ and $\nu_{\text{eff}}$ following, e.g., Grosvenor et al. (2018) and Sinclair et al. (2019) as follows:

$$N_d = \frac{\sqrt{5}}{2\pi k} \left( \frac{f_{\text{ad}} c_w \tau_c}{Q_{\text{ext}} \rho_w R_{\text{eff}}^5} \right)^{\frac{1}{2}} \qquad (3)$$

where $f_{\text{ad}}$ is the subadiabatic factor, $\tau_c$ is the cloud optical depth, $c_w$ is the adiabatic condensation rate (e.g., Brenguier et al., 2000; Painemal and Zuidema, 2011), $Q_{\text{ext}}$ is a unitless extinction efficiency factor, $\rho_w$ is the bulk density of liquid water (1000 kg m$^{-3}$), and $k$ is a parameter that relates $\nu_{\text{eff}}$ to $R_{\text{eff}}$ following:

$$k = \left( \frac{R_v}{R_{\text{eff}}} \right)^3 = (1 - \nu_{\text{eff}})(1 - 2\nu_{\text{eff}}) \qquad (4)$$

where $R_v$ is the volume-mean droplet radius (Grosvenor et al., 2018). Here, $k$, $R_{\text{eff}}$, and $\tau_c$ are retrieved by the RSP, $Q_{\text{ext}}$ is assumed to be 2 for Mie scattering, $c_w$ is calculated via dropsondes from the CAMP$^2$Ex flight and decreases with altitude, and $f_{\text{ad}}$ is assumed to be 0.8 following in situ profiles from the NASA North Atlantic Aerosols and Marine Ecosystems Study (Behrenfeld et al., 2019; Alexandrov et al., 2018). Grosvenor et al. (2018) conclude that different observations suggest $f_{\text{ad}}$ of 0.66 ± 0.22, which encompasses the value chosen here. However, since $N_d$ scales with $f_{ad}^{\frac{1}{2}}$, using 0.66 versus 0.8 would

decrease $N_d$ by only a factor of 0.9. RSP retrievals are considered to represent $\sim 1$ optical depth below cloud top (Miller et al., 2018; Alexandrov et al., 2018). Importantly, RSP retrievals are inherently truncated at a size limit that is dependent on the size distribution being observed (see Fig. 6 in Reid et al., 2023). Here, we assume this size limit be a diameter (radius) of 200 (100) $\mu$m. Using in situ cloud microphysics (discussed next), we explore the implications of using RSP retrievals at this limit and discuss relevant sensitivities.

**2.3 Clouds: In situ cloud microphysics measurements**

In situ cloud probes are used to evaluate DSD characteristics within cloud cores. Probes were instrumented on both aircraft but only Learjet measurements are used herein. These include the fast forward scattering spectrometer probe (FFSSP; O'Connor et al., 2008; Lawson et al., 2017), the Nevzorov liquid and total water content device (Korolev et al., 1998), the 10-$\mu$m channel 2D-stereo (2D-S) optical array probe (Lawson et al., 2006), and the high-volume precipitation spectrometer probe (HVPS;

Lawson et al., 1998). In situ probes are used to evaluate similarities with RSP retrievals and to provide continuous size distributions that extend beyond the limits of the RSP. Further discussion of combining these instruments to produce continuous size distributions is provided in Appendix D.

**2.4 Case Description**

The event on 25 September 2019 corresponded to Research Flight 14 (RF14) for the P-3B and RF12 for the SPEC Learjet

from $\sim$ 0300 UTC to 0900 UTC. The 0 °C level during this case was $\sim$ 5 km above ground level (AGL). The P-3B aircraft initially sampled cumulus clusters with CTHs $\sim$ 4-5 km, as demonstrated in Fig. 2, which shows a 30-km overpass of a cumulus congestus cluster during this case along with time-height cross sections of radar reflectivity from the W-band 3$^{\text{rd}}$



Generation Airborne Precipitation Radar (APR3) and a time-series of RSP-retrieved cloud-top $N_d$ and $R_{eff}$. Cloud-top heights over ocean were likely limited by a prominent dry layer $\sim$ 4-4.5 km AGL where a dewpoint depression of $\sim$ 20 °C was evident

in dropsondes (see Fig. 3b). However, the strength of imposed large-scale vertical motion additionally modulated maximum CTHs in simulations, which is connected to the thermodynamic profile by shifting the vertical structure of relative humidity.

The P-3B mostly sampled at a cruising altitude of $\sim$ 7 km AGL. The overpass shown in Fig. 2 was over ocean and during the earlier portion of the flight. During the latter portion of the flight (after $\sim$ 0600 UTC), the P-3B sampled a more vigorous cluster of convection near the southern end of Catanduanes Island (see location in Fig. 4). Continental surface fluxes and

enhanced terrain (up to $\sim$ 1.5 km above sea level) from the island are presumed to have invigorated convection along with continued moistening from the cloud system situated to the north of the island (not shown). Importantly, sampling of clouds over Catanduanes Island by the P-3B were mostly in-situ, which since the RSP requires the plane to fly above cloud top, means the RSP was unable to sample the tops of the most vigorous clouds. Therefore, forthcoming RSP retrievals are shown mostly for cumulus congestus clusters over ocean with CTHs warmer than 0 °C before the aircraft sampled the more vigorous

convection over the island.

The Learjet also performed a number of cloud penetrations prior to 0600 UTC, after which it continuously sampled in-cloud transects at higher altitudes over the southern portion of Catanduanes Island (Fig. 4). While sampling the island convection, the Learjet indeed sampled CTHs reaching up to 7 km AGL where ice formation proceeded. In-situ images from a cloud particle imager (CPI; Lawson et al., 2001; Woods et al., 2018) showed indications of ice multiplication by means of fractured drops and

spicules. Ice multiplication for this event, including the development of a drop shattering parameterization based on laboratory data, is explored in a subsequent study.

## 3    Model Setup and Experimental Design

Here we describe the LES model setup and experimental design. The LES is initialized with horizontally uniform thermodynamic and large-scale vertical motion ($w_{LS}$) profiles. While thermodynamic profiles can be obtained from dropsondes, aircraft,

and/or radiosondes released nearby, $w_{LS}$ is more difficult to quantify observationally. Prior work indicated the importance of including $w_{LS}$ as LES forcing in order to realize observed conditions. In the absence of observational data, we use mesoscale simulations by the NASA Unified Weather Research and Forecasting (NU-WRF; Peters-Lidard et al., 2015) model to estimate thermodynamic and dynamic forcing.

### 3.1    Harvesting Thermodynamic and Dynamic Forcing from WRF

Specific details of the NU-WRF simulation set-up are provided in Appendix A, and only a brief description is provided here. We employ a nested set-up with an outer domain with a horizontal mesh of 3 km and an inner domain with a horizontal mesh of 600 m. The inner 600-m mesh domain is shown in Fig. 4 which encompasses the relevant portions of the P-3B and SPEC Learjet flights. Thermodynamic profiles (temperature and humidity) are averaged over this domain and shown in Fig. 3b. Two dropsondes from the flight are also shown in Fig. 3b, where dropsonde #1 was released in an ambient environment far from



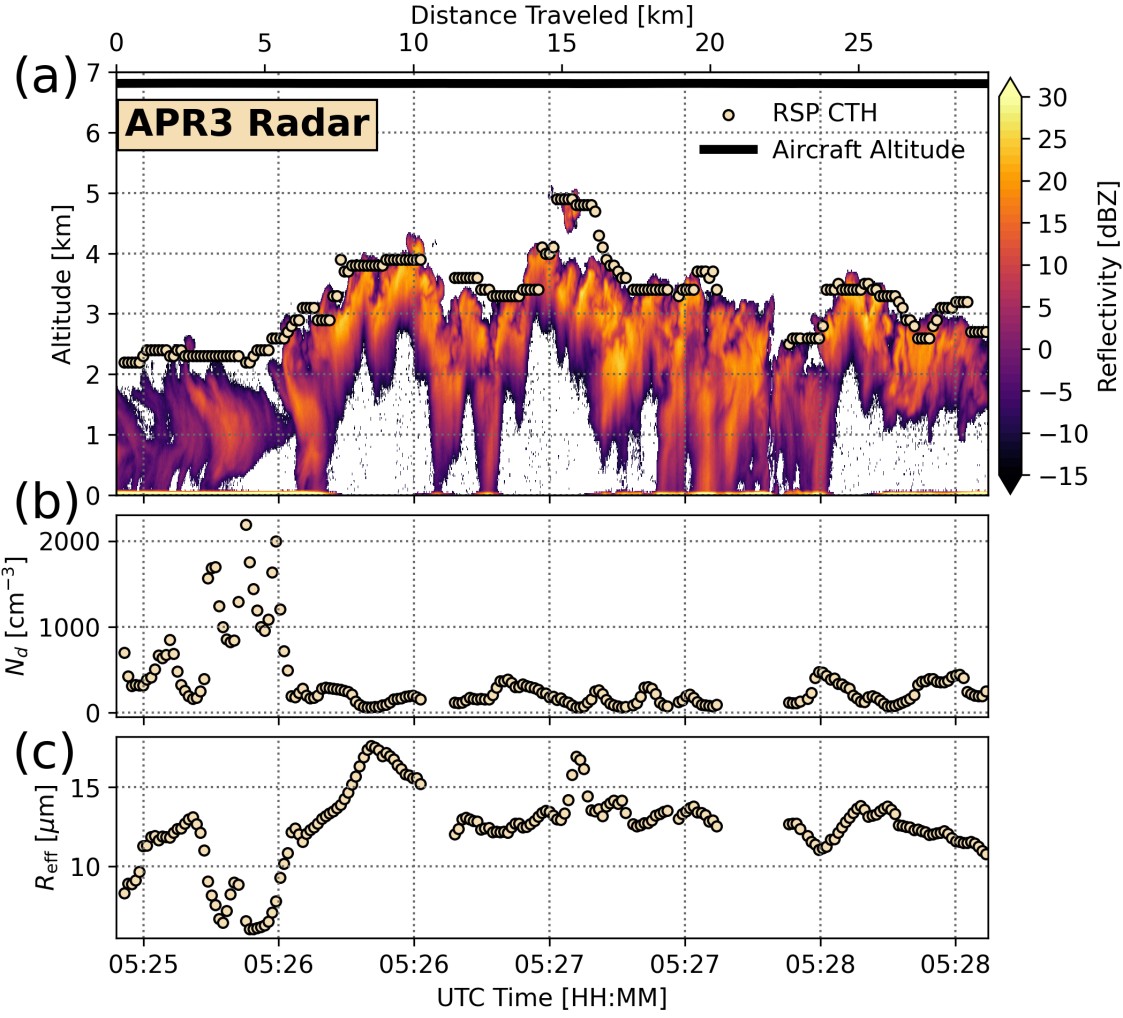

**Figure 2.** Time series of (a) profiles of W-band radar reflectivity from the 3[rd] Generation Airborne Precipitation Radar (APR3) and CTHs retrieved from the Research Scanning Polarimeter (RSP), (b) the RSP-retrieved cloud-top drop number concentration ($N_d$), and (c) the RSP-retrieved cloud-top drop effective radius ($R_{\text{eff}}$).



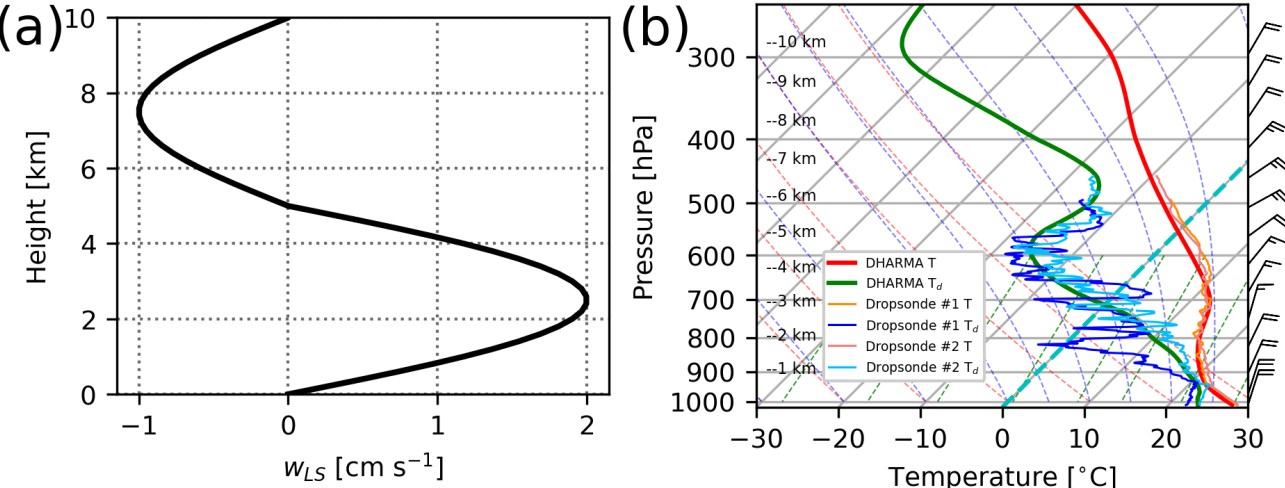

**Figure 3.** Profiles of (a) large-scale vertical motion ($w_{LS}$) and (b) thermodynamic profiles shown on a skew T – log P diagram from the NU-WRF-derived sounding used to initialize the LES and two dropsondes released in the ambient (#1) and near-convection (#2) environments by the P-3B aircraft, as included in the legend.

active convection and dropsonde #2 was released in an ambient but near-convection environment. Regardless, both dropsondes agree well with the profile derived from the NU-WRF domain-average.

Various sectors of the domain shown in Fig. 4 were evaluated to derive $w_{LS}$. While there was considerable variability across the domain and time, spatial and temporal averaging (across hours that corresponded to flight times, $\sim$ 0300-0900 UTC) yielded a characteristic profile that is shown in Fig. 3a, with positive vertical motion in the lowest 5 km and subsidence between 5 km and 10 km. The subsidence between 5 and 10 km conceptually seems consistent with the extreme dry layer represented in the sounding $\sim$ 5 km. As a result, we proceed with the idealized $w_{LS}$ profile provided in Fig. 3a that represents a vertical mean $w_{LS}$ of 0.31 cm s$^{-1}$. We note that while we do not consider this $w_{LS}$ profile to be very well-constrained with respect to observations, we do consider it to be plausible and sufficient for representing the large-scale vertical motion in the presented simulations. Furthermore, because a foundational objective of this study is to establish a cumulus congestus case study that can be evaluated in SCM simulations for convective microphysics development in large-scale models, it is pertinent to provide a measure of $w_{LS}$ to also force the SCM. Appendix B provides further details of the importance of including $w_{LS}$ to force the LES dynamic conditions.

## 3.2 Large Eddy Simulations

Large eddy simulations are performed using the Distributed Hydrodynamic Aerosol and Radiative Modeling for Atmospheres (DHARMA; Stevens and Bretherton, 1996; Stevens et al., 2002; Ackerman et al., 2000) model with doubly periodic boundary



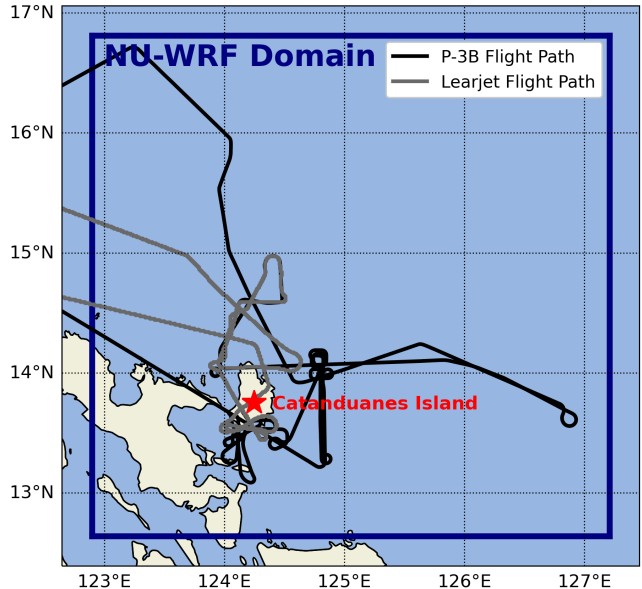

**Figure 4.** Inner NU-WRF domain used for harvesting large-scale vertical motion and thermodynamics to initialize the DHARMA simulations (blue box) and flight paths of the SPEC Learjet and the NASA P-3B aircraft.

conditions and a horizontal mesh at 100 m. The domain size is 19.2 x 19.2 km and spans 20 km vertically. Vertical grid spacing increases linearly from ∼20-100 m in the lowest 1 km, constant at 100 m between 1 and 15 km, and coarsens above 15 km. A dynamic Smagorinsky turbulence model from Kirkpatrick et al. (2006) is used for parameterizing subgrid fluxes, and ocean surface fluxes follow the bulk aerodynamic formula of Zeng et al. (1998). Radiation is neglected for simplicity because the

impacts of radiative heating and cooling on microphysical processes are considered to be second-order relative to congestus dynamics. Simulations are integrated for 12 h and use a dynamics time step of 1 second. Although ice particles were identified at temperatures below 0 °C when the aircrafts sampled invigorated convection over the southern portion of Catanduanes Island, we neglect ice here to narrow focus on warm-phase microphysical processes. The inclusion of ice will be discussed in a follow-up study, but sensitivity tests including ice indicate this does not significantly affect the warm-phase microphysical evolution

described here. Simulations are performed using both bulk and bin (size-resolved) microphysics schemes. An overview of all sensitivity experiments is provided in Table 1.

### 3.2.1   Bulk Microphysics

Bulk microphysics follow a substantially modified version of the two-moment scheme of Morrison et al. (2005, 2009) using only cloud and rain hydrometeor species, with the addition of three aerosol modes. The control simulation (CNTL) and all but

one sensitivity simulation use the droplet autoconversion and self-collection parameterization of Seifert and Beheng (2001). Rain accretion, self-collection, breakup, and fall speed follow Seifert (2008), and the gamma size distribution for rain uses a



shape parameter of 3. Sensitivity to the warm-rain formulation is tested using the Khairoutdinov and Kogan (2000) scheme for autoconversion and accretion and employing an exponential size distribution to emulate the parameterization most commonly employed in bulk microphysics models (experiment named KK). Aerosol is activated using a prognostic supersaturation value after microphysical relaxation that follows from Morrison and Grabowski (2008a). The number concentration of activated and unactivated aerosol are tracked throughout model integration. The sensitivity of $N_d$ and $R_{eff}$ profile modulation to collision-coalescence is examined by turning off autoconversion as the lower extreme (experiment named NO_AC) and by scaling the autoconversion efficiency by a factor of two as an upper extreme (experiment named 2X_AC). The impact of the height-resolved aerosol PSD is explored by fixing each mode of the trimodal lognormal size distributions to the cloud-base value, therefore making the profile constant with height as a number mixing ratio (experiment named FIXED_AERO). The combined sensitivity to autoconversion, collision-coalescence, and the height-resolved aerosol PSD is then explored in the FIXED_AERO_NO_AC experiment. Finally, a resolution sensitivity test is performed with the same physics as CNTL but with a horizontal mesh at 50 m that is discussed in Appendix C.

### 3.2.2  Bin Microphysics

The bin microphysics scheme is based on the Community Aerosol-Radiation-Microphysics Application (CARMA) code (Ackerman et al., 1995; Jensen et al., 1998). Prognostic species include unactivated aerosol number in three modes, liquid drop number, and aerosol core mass in liquid drops, the last of which is not considered in detail here. For each species, we use a geometric mass grid with a constant ratio of masses between two adjacent bins. The mass ratio ($\mathcal{M}$) is defined as $m_{i+1}/m_i = 2^{1/s}$, where $m_i$ is the drop mass in the $i^{th}$ bin and $s$ is a bin width parameter such that drop mass is doubled every $s$ bin. For liquid drops, $\mathcal{M} = 1.65$ ($s \sim 1.384$) and for aerosols $\mathcal{M} = 1.35$ ($s \sim 2.309$). For spherical particles, the corresponding geometric size grid is $r_{i+1}/r_i = 2^{1/(3s)}$ where $r_i$ is the radius of the $i^{th}$ bin. For aerosols, the mass in the smallest bin corresponds to a diameter of $\sim 10$ nm and the largest bin corresponds to a diameter of $\sim 1.5$ $\mu$m. For liquid drops, the size bins range from 2 $\mu$m to 7.78 mm in diameter. Lee et al. (2021) evaluated the impacts of numerical broadening in CARMA using the geometric grid compared to a hybrid grid that transitions from linear to geometric bin spacing, with the linear grid intended to limit numerical broadening caused by solving condensation. In parcel simulations, Lee et al. (2021) found that the introduction of turbulent-induced collision enhancement (discussed below) limited the effects of numerical broadening from condensation alone and reduced differences between the two grid choices relative to the absence of turbulent enhancement. Furthermore, in LES of a drizzling marine stratocumulus case study, they found that the hybrid grid led to smaller drops and delayed onset of surface precipitation that agreed slightly better with radar observations of DSD moments, but that overall differences between simulations were small relative to differences with observations. As will be shown, bin simulations herein consistently produced DSDs that were too narrow, thereby motivating retention of the geometric grid in order to not inhibit precipitation formation.

Specific details of microphysical processes including condensation, evaporation, and sedimentation are given in Ackerman et al. (1995). As in the bulk simulations, we employ a trimodal aerosol distribution (Fig. 1) with a hygroscopicity parameter of 0.4 for all modes. Aerosol activation follows from Ackerman et al. (1995). Collision-coalescence is performed using the





**Table 1.** Simulation Details. Line colors devoted to each simulation in forthcoming plots are given in the rightmost column.

| Simulation Name | Description | Line Color |
|---|---|---|
| CNTL | Setup described in text w/ Seifert and Beheng (2001) warm-rain formulation | Gray |
| NO_AC | As in CNTL, w/ autoconversion turned off | Red |
| FIXED_AERO | As in CNTL, w/ aerosol PSD fixed to cloud-base value throughout profile | Orange |
| FIXED_AERO_NO_AC | As in CNTL, w/ autoconversion turned off and aerosol PSD fixed to cloud-base value throughout profile | Purple |
| 2X_AC | As in CNTL, w/ autoconversion efficiency scaled by a factor of 2 | Blue |
| KK | As in CNTL, w/ Khairoutdinov and Kogan (2000) warm-rain formulation and an exponential size distribution | Green |
| BIN | As in CNTL, w/ size-resolved bin microphysics | Pink |
| BIN_TURB | As in BIN, w/ turbulent enhancement of collision-coalescence | Brown |
| BIN_TURB_10X | As in BIN_TURB, w/ turbulent enhancement scaled by a factor of 10 | Light Blue |

exponential collection scheme of Bott (2000) to solve the stochastic collection equation with collision efficiencies from Hall (1980). Raindrop breakup follows from Hall (1980) and Low and List (1982). The baseline bin microphysics simulation is named BIN. We perform two additional sensitivity experiments to explore the role of turbulent enhancement of collision-coalescence. In the first experiment (BIN_TURB), the turbulent collision kernel from Ayala et al. (2008) and Grabowski and Wang (2013) is incorporated following the implementation described by Lee et al. (2021) that uses the explicitly calculated turbulent kinetic energy dissipation rate ($\epsilon$) from the subgrid-scale (SGS) diffusion scheme. Chen et al. (2018) found in large eddy simulations of an Arctic mixed-phase cloud that the modeled turbulent broadening was narrower than observations of Doppler spectral width. They calculated that the total dissipation rate (i.e., numerical plus SGS) was a factor of 6 larger than the SGS dissipation rate and agreed better with observations. Moreover, Chen et al. (2018) revisited the study of Rémillard et al. (2017) who simulated a drizzling marine stratocumulus case and determined that the total dissipation rate was $\sim 3$ times larger than the SGS dissipation rate, demonstrating that the the magnitude of the ratio between the total and SGS dissipation rate is strongly case-dependent. Here, we calculate a total dissipation rate $\sim 10$ times larger than the SGS scheme (not shown), suggesting the more turbulent cloud environment simulated herein experiences substantially greater turbulent broadening than that calculated by the SGS scheme. Therefore, in a second experiment, $\epsilon$ is scaled by a factor of 10 (BIN_TURB_10X), which conceptually can be considered the most appropriate parameterization using the bin scheme with turbulent enhancement included.

### 3.3 Thermal Identification and Tracking Framework

To investigate the role of entrainment in modulating profiles of $N_d$ and to facilitate interpretation of cloud droplet production and evolution within their source elements (i.e., cumulus thermals), a thermal identification and tracking framework is



employed. The algorithm used is described in Hernandez-Deckers and Sherwood (2016), based on an early version by Sher-
wood et al. (2013), and has been demonstrated in several recent studies to investigate the interacting roles of microphysics,
aerosols, and convective dynamics (e.g., Hernandez-Deckers and Sherwood, 2018; Hernandez-Deckers et al., 2022; Matsui
et al., 2024). A thorough description of the algorithm is provided in Hernandez-Deckers et al. (2022) and Matsui et al. (2024)
so only a brief description is provided here. Rising volumes of cloudy air, above a minimum vertical velocity ($w$) threshold (1

295  m s$^{-1}$) and condensate threshold (0.01 g kg$^{-1}$), are tracked at high temporal frequency (1 minute). The minimum lifetime for
thermals identified in this study is 3 minutes. Composite thermal statistics are centered around a thermal's maximum ascent
rate (which differs from the maximum $w$ due to inhomogeneity across the thermal's width resulting from toroidal circulations).
The algorithm assumes a spherical shape, which Hernandez-Deckers and Sherwood (2016) showed to be a valid approximation
compared to more plume-like structures. In transient convection simulations, Sherwood et al. (2013) and Hernandez-Deckers

and Sherwood (2016) showed that thermals are generally short-lived ($\sim$ 4-5 minutes) and rather small (relative to the horizontal
mesh we use), which is consistent with thermals detected in the current study (not shown).

## 4  Results

### 4.1  Observed Profiles of $N_d$ and $R_{\text{eff}}$

Cloud-top $N_d$ and $R_{\text{eff}}$ retrieved by the RSP are composited over temperatures warmer than 0 °C and sorted as a function of

CTH (Fig. 5). In situ data from the Learjet FFSSP are also shown in Fig. 5 for points with liquid water content (LWC) > 0.1 g
m$^3$. Median $N_d$ from the RSP decrease from $\sim$ 500 cm$^{-3}$ for CTHs between 0 and 1 km AGL to $\sim$ 100 cm$^{-3}$ for CTHs between
4 and 5 km AGL (Fig. 5a). Drop effective radius follows an inverse relationship with $N_d$, increasing from $\sim$ 5 $\mu$m for CTHs
below 1 km AGL to $\sim$ 14 $\mu$m for CTHs between 4 and 5 km AGL. Notably, $R_{\text{eff}}$ values of up to 14 $\mu$m are conceptually aligned
with the onset of precipitation at around this altitude (Rosenfeld and Gutman, 1994; Gerber, 1996; Andreae et al., 2004; Freud

and Rosenfeld, 2012). Active precipitation in this case is consistent with the radar reflectivity transect shown in Fig. 2a with
relatively high reflectivity values (> 20 dBZ) near cloud top and vertically continuous radar echoes reaching the surface.

In situ data from the FFSSP show smaller $N_d$ relative to RSP for each altitude bin by around a factor of $\sim$ 1.5, while FFSSP
$R_{\text{eff}}$ is slightly larger than the RSP. We note that the largest size bin for the FFSSP is 50 $\mu$m in diameter, while we are here
assuming that the RSP DSD is truncated at a radius of 100 $\mu$m (diameter of 200 $\mu$m). While it is possible to merge the FFSSP

DSD with the 2D-S10 to get an extended distribution, doing so requires robust statistical averaging for stitching the DSDs
together. Here, we chose to retain the 1-Hz native resolution of the FFSSP. This is justified with the knowledge that (1) $N_d$
would be largely insensitive to the inclusion of larger particles, (2) $R_{\text{eff}}$ would only increase by including an instrument with
larger size bins, and (3) the in situ $R_{\text{eff}}$ is already larger than the RSP. We also note that using a smaller LWC threshold of 0.01
g m$^3$ decreases the in situ $N_d$, shifting it further from the RSP.

Discrepancies between RSP and in situ measurements may be due to several factors, besides at the foundation that RSP
retrieves cloud top while in situ measurements are ideally in and near the cloud core. For $N_d$, the RSP retrieval is limited by
assumptions in Eq. 3, which is derived from an adiabatic cloud model (e.g., Grosvenor et al., 2018). This method assumes





that $N_d$ is constant with height for a given cloud profile and that LWC increases linearly with height as a constant fraction of its adiabatic value ($f_{ad}$). The presented RSP-retrieved $N_d$ assumes $f_{ad} = 0.8$, which may be too high for these cumulus clouds

that are subject to substantial entrainment. However, $f_{ad}$ can also be highly variable in addition to its uncertainty. Ultimately, Eq. 3 is more appropriate for stratocumulus clouds than for cumulus. A lower value of $f_{ad}$ would decrease $N_d$, which scales with the square root of $f_{ad}$. Moreover, errors in $R_{eff}$ propagate to errors in $N_d$ that scale with the power of -5/2. Therefore, slightly lower RSP $R_{eff}$ relative to in situ is conceptually consistent with high RSP $N_d$. We note that for the majority of height bins, the in situ $R_{eff}$ lies within the RSP's interquartile range. Nonetheless, we accept these uncertainties here to explore the

ability for simulations to reproduce profiles of $N_d$ and $R_{eff}$ that are within the range provided by the RSP retrievals and in situ measurements.

## 4.2 Simulated Cloud System Evolution

We represent the simulated cloud-system evolution via time-height series of domain-averaged in-cloud cloud droplet number concentrations ($N_c$; Fig. 6), which excludes the rain species in order to better emphasize the evolution of cloud droplets only.

Clouds initiate within the first 30 minutes and maintain a cloud base height (CBH) of $\sim 0.5$ km AGL throughout the entirety of the simulation. Cloud tops grow monotonically from $\sim 2$ km initially to $\sim 6$ km at $\sim 10$ h for CNTL. Time series of rain water path (RWP) are also shown in Fig. 6, where rain onset occurs at $\sim 9$ h in CNTL. Notably, the CTH stabilizes after sufficient production of precipitation. As discussed later, the simulated system is composed of numerous thermals that successively reach higher altitudes. After the onset of precipitation in CNTL, cold pools form such that the system begins to cluster with relatively

more vigorous convection (not shown).

Cloud droplet number concentrations decrease with height in CNTL from $\sim 500$-$600$ cm$^{-3}$ near cloud base down to $\sim 100$ cm$^{-3}$ near cloud top in the last 3 hours. In comparison, $N_c$ in NO_AC (i.e., no rain formation, Fig. 6d) decreases with height to only $\sim 200$-$300$ cm$^{-3}$ near the end of the simulation, indicating the role of collision-coalescence in scavenging $N_c$ in CNTL. The FIXED_AERO simulation (Fig. 6b) realizes higher $N_c$ throughout the entire profile, which results mainly from the 3$^{rd}$ and

largest mode of the trimodal $N_a$ distribution (Fig. 1). The largest mode is activated most frequently (not shown), and by forcing a constant profile with a value equal to that at cloud base ($\sim 650$ cm$^{-3}$; Fig. 1a) instead of decreasing with altitude, it results in an effectively more polluted profile relative to the CNTL profile. This also acts to slightly delay higher RWPs relative to CNTL and leads to slightly weaker $N_c$ scavenging. The FIXED_AERO_NO_AC simulation (Fig. 6d) represents the most extreme difference relative to CNTL, with $N_c$ only decreasing by $\sim 200$ cm$^{-3}$ between cloud base and cloud top near the end of the

simulation, and even more modest decreases at times prior. The KK simulation experiences an earlier onset of precipitation that realizes significantly reduced $N_c$ by hour 12 (Fig. 6d). The KK scheme has a more aggressive accretion process than the Seifert and Beheng (2001) scheme (Stevens and Seifert, 2008) and this experiment uses an exponential size distribution (as opposed to a gamma with a shape parameter of 3 as in all other experiments). The combination of these parameter choices leads to reduced RWP relative to CNTL and larger surface precipitation rates (not shown). Likewise, precipitation onset occurs earlier

in 2X_AC (Fig. 6f) when autoconversion is scaled higher, which leads to the most significant scavenging of $N_c$ by precipitation and a sharper decay of the system.



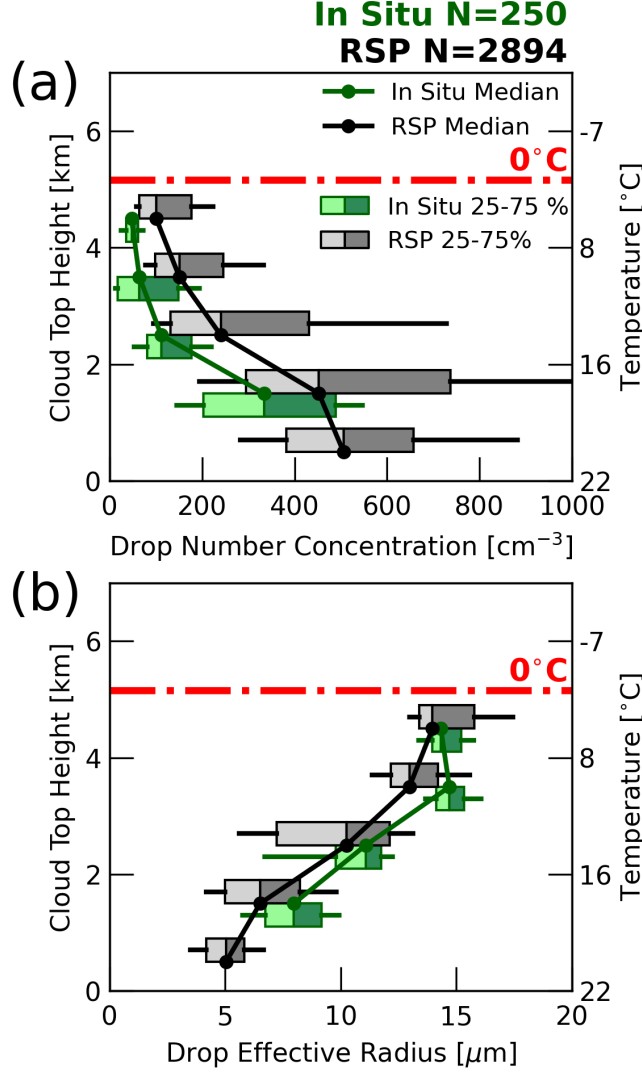

**Figure 5.** Profiles of drop (a) number concentration ($N_d$) and (b) effective radius ($R_{eff}$) as a function of cloud top height (CTH) bins (bin width = 1 km) from the RSP (grays) and as a function of in-situ altitude bins from the FFSSP (greens). Boxes show the interquartile range and whiskers show the 10[th] and 90[th] percentiles. The RSP and in situ data are offset from the center of the 1-km bins with RSP on top and in situ on bottom. Circle markers with lines indicate the median and are placed at the center of the height bin for both instruments. Only bins with at least 10 data points are shown for each instrument.





**Figure 6.** Time-height series of in-cloud domain average cloud droplet number concentration ($N_c$) for the (a) CNTL, (b) FIXED_AERO, (c) KK, (d) NO_AC, (e) FIXED_AERO_NO_AC, (f) 2X_AC, (g) BIN, (h) BIN_TURB, and (i) BIN_TURB_10X simulations. Blue lines (right ordinates) show the rain water path (RWP) time series.



For the purpose of discussing system evolution using the bin scheme in comparison to the bulk scheme, liquid water in the bin scheme is arbitrarily separated using a radius threshold of 25 $\mu$m to categorize as cloud and rain species. However, we note that further analysis herein will make no such separation (unless otherwise stated). There are notable differences between the

BIN experiment (Fig. 6g) and CNTL. Cloud droplet number concentrations are higher in BIN throughout the entire profile, indicating more efficient aerosol activation in BIN rleative to CNTL. Moreover, very minimal precipitation-sized drops are present to have any significant scavenging effect on $N_c$. Indeed, by the end of the simulation, the domain in BIN is covered in an outflow cloud layer between $\sim$ 5-6 km, with no cold pool formation or clustering of cumulus clouds. Turbulent enhancement of collision-coalescence (BIN_TURB; Fig. 6h) results in slightly more precipitation that decreases $N_c$ slightly above 4 km near

the end of the simulation. Interestingly, scaling turbulent enhancement by a factor of 10 (BIN_TURB_10X; Fig. 6i) does enhance precipitation further and scavenges smaller drops more efficiently, but still not to the degree realized by CNTL. One potential cause for delayed onset and weaker precipitation in the bin scheme may be due to more efficient aerosol activation that yields the effect of a more polluted environment, though the impact of the collision kernel cannot be negated.

### 4.3 Comparison of Simulations and RSP Retrievals

Simulated CTH is calculated by integrating optical depth ($\tau$) from the top of the domain downward until it exceeds a threshold value of 1 (to match the assumed $\tau$ threshold of RSP), where $\tau(z)$ is calculated following, for example, Hansen and Travis (1974) and Stephens (1978):

$$\tau(z) = \int_{z}^{z+\Delta z} \beta_{\text{ext}} dz = \int_{z}^{z+\Delta z} \int_{0}^{\infty} \pi Q_{\text{ext}} \frac{dn}{dr} r^2 dr dz \tag{5}$$

where $\beta_{\text{ext}}$ is the extinction coefficient, $Q_{\text{ext}}$ is the dimensionless extinction efficiency (assumed to be 2), $r$ is drop radius, $dz$ is the height difference across a given level, and $dn/dr$ is the size-resolved drop number distribution for a given height. For the

bulk scheme, $dn/dr$ is reconstructed using the gamma distribution parameters and includes contributions from both cloud and rain species. The range of drop radii is chosen to correspond to the 50 bins used by the bin scheme (see Section 3.2.2). Drop number concentration is the sum of the cloud and rain species in the bulk scheme. Effective radius is calculated in the typical way as:

$$R_{\text{eff}} = \frac{\int_{0}^{100\mu m} \frac{dn}{dr} r^3 dr}{\int_{0}^{100\mu m} \frac{dn}{dr} r^2 dr} \tag{6}$$

where for the bulk scheme, $dn/dr$ is the sum of drops from the cloud and rain species for a given discretized bin and is again

reconstructed from their respective gamma distribution parameters. Note $dn/dr$ is only integrated to a size limit of $r$ = 100 $\mu$m since the RSP is not sensitive to larger drops. Implications for this truncation size are discussed below. Finally, cloud-top $N_d$ and $R_{\text{eff}}$ are calculated using an extinction-weighting following:

$$N_d^{\text{top}} = \int_{z(\tau>0)}^{z(\tau\geq1)} N_d \beta_{\text{ext}} dz \left/ \int_{z(\tau>0)}^{z(\tau\geq1)} \beta_{\text{ext}} dz \right. \tag{7}$$





$$R_{\text{eff}}^{\text{top}} = \frac{\int\limits_{z(\tau>0)}^{z(\tau\geq1)} \int\limits_{0}^{100\mu m} \frac{dn}{dr} r^3 \beta_{\text{ext}} dr dz \Big/ \int\limits_{z(\tau>0)}^{z(\tau\geq1)} \beta_{\text{ext}} dz}{\int\limits_{z(\tau>0)}^{z(\tau\geq1)} \int\limits_{0}^{100\mu m} \frac{dn}{dr} r^2 \beta_{\text{ext}} dr dz \Big/ \int\limits_{z(\tau>0)}^{z(\tau\geq1)} \beta_{\text{ext}} dz}. \tag{8}$$

where integration of each size distribution moment is performed to an upper radius size limit of 100 $\mu$m. Equations 7 and 8 represent the weighting between diffuse cloud top and a typical step increase in optical depth across the layer we define

explicitly as cloud top (i.e., when accumulated $\tau$ exceeds a value of 1 from domain-top), which is consistent with the RSP retrievals (e.g., Alexandrov et al., 2012a).

Profile distributions of cloud-top $N_d$ and $R_{\text{eff}}$ for the CNTL and BIN_TURB_10X simulations are shown in Fig. 7, with the latter chosen as it represents the most appropriate implementation of turbulence-enhanced collision-coalescence with the DHARMA model and the most efficient precipitation formation among the three bin experiments. Cloud-top statistics are

accumulated across the last three hours of the simulation to capture the effect of precipitation that was active in the observed system (see Fig. 2a), and to roughly represent flight timing that corresponds to initialization time of the NU-WRF mesoscale simulation used for DHARMA initial conditions. The CNTL simulation reasonably represents the decrease of cloud-top $N_d$ with height (Fig. 7a) with the maximum difference relative to RSP occurring for CTHs in the lowest 1 km, where simulated cloud-top $N_d$ is slightly lower, but only by $\sim$ 50 cm$^{-3}$ in the median. Simulated $R_{\text{eff}}$ in CNTL (Fig. 7c) is consistently lower

than the RSP at all CTHs, with differences increasing with increasing CTH. The BIN_TURB_10X scheme notably produces wider distributions of cloud-top $N_d$ at any given CTH relative to CNTL (Fig. 7b), but does not capture the correct structure of the cloud-top $N_d$ profile relative to RSP. While the median cloud-top $N_d$ in the lowest 1-km agrees with RSP, the median profile decreases too sharply with height, though we note that $N_d$ values between $\sim$ 2-3 km are closer to the in situ measurements. Cloud-top $R_{\text{eff}}$ in BIN_TURB_10X (Fig. 7d) agrees well with RSP below 2 km, but diverges similarly to CNTL at higher

CTHs. We note that qualitative results from Fig. 7 are not sensitive to the size threshold used to calculate $N_d$ or $R_{\text{eff}}$ and in fact are very similar even if calculated using the entire size distribution, which is likely because of consistent DSD characteristics at cloud-top as opposed to an evolving distribution in the source thermals. Although the RSP did not sample cloud-tops for this case at sub-0 °C temperatures, the simulations and 1-Hz FFSSP samples imply continued trends of both $N_d$ and $R_{\text{eff}}$ at colder temperatures.

Median profiles of cloud-top $N_d$ and $R_{\text{eff}}$ for all simulations are shown in Fig. 8, with the inclusion of $N_d$ in units of number mixing ratio (kg$^{-1}$) in Fig. 8b,e. This air density-dependent conversion is performed to control for the impacts of dilution by expansion, which Morrison et al. (2022) showed to account for 41% of $N_d$ reduction between 4 and 9 km in a high-based congestus simulation. In doing so, we isolate the relative impacts on $N_d$ via collision-coalescence and aerosol profile representation using the sensitivity experiments, with any remaining changes in the profile attributed to dilution by entrainment or,

conversely, subsequent activation of cloud droplets above cloud base. For the bulk simulations (Fig. 8b), cloud-top $N_d$ follows an intuitive pattern. The 2X_AC and KK simulations produce a sharper reduction of $N_d$ with height due to more aggressive



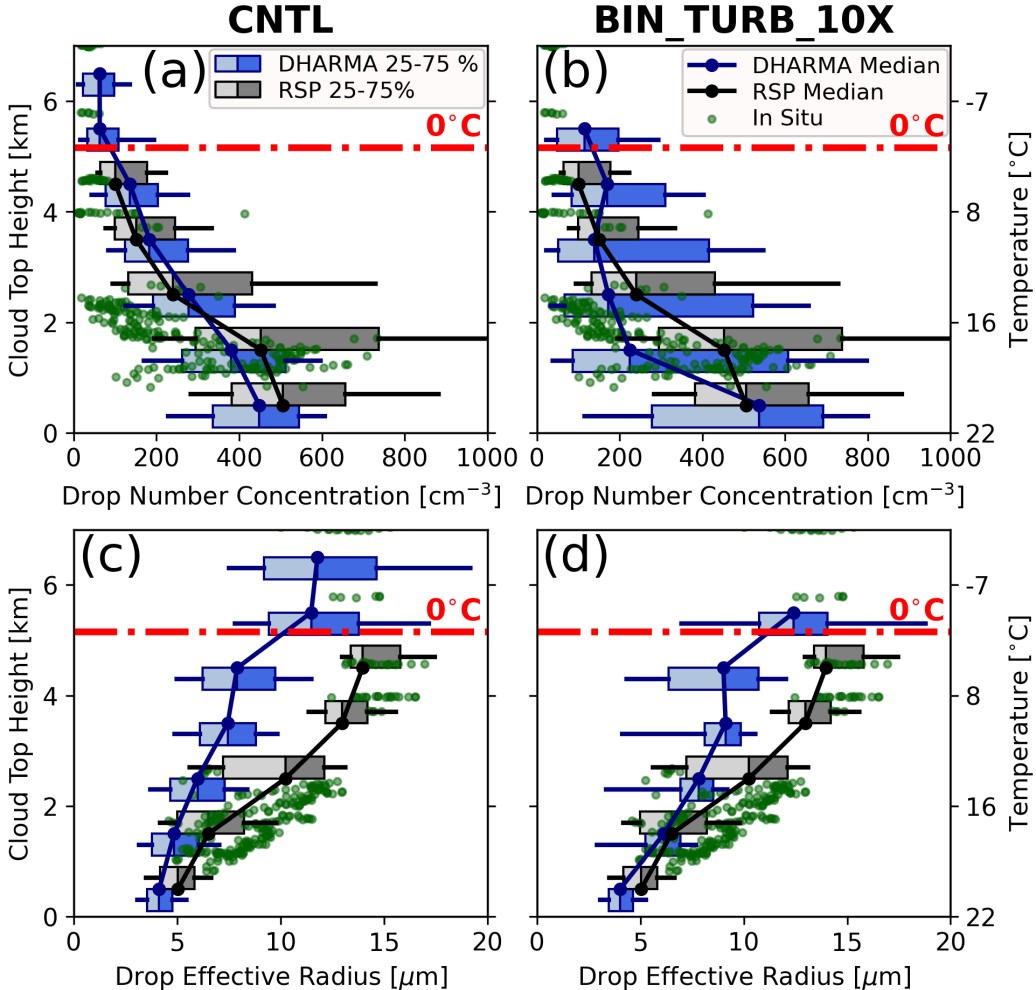

**Figure 7.** Profiles of cloud-top (a,b) drop number concentration and (c,d) drop effective radius for the CNTL simulation (left column) and the BIN_TURB_10X simulation (right column) as a function of cloud top height. Individual 1-Hz in situ measurements from the FFSSP are also shown as green dots. Lines connected by circle markers are the medians for simulations and the RSP. Boxes show the interquartile range and whiskers show the 10[th] and 90[th] percentiles. Distributions from simulations are across the entire domain for the last 3 h of the simulation. The height bin width is 1 km with the RSP and simulations offset relative to the mid-bin for visibility.





precipitation formation. Conversely, the NO_AC simulation produces a much shallower decrease due to no precipitation, while FIXED_AERO produces a shallower slope due to inefficient precipitation owing to the effectively more polluted profile. At the extreme, the FIXED_AERO_NO_AC profile shows no significant reduction of $N_d$ at all. Profiles of cloud-top $R_{\text{eff}}$ (Fig. 8c) for
the sensitivity simulations generally follow an inverse relationship from $N_d$ relative to CNTL. That is, KK and 2X_AC produce larger $R_{\text{eff}}$ while NO_AC, FIXED_AERO, and FIXED_AERO_NO_AC produce smaller $R_{\text{eff}}$ compared to CNTL. Notably, every bulk simulation produces lower cloud-top $R_{\text{eff}}$ compared to RSP. Each of the bin simulations produce lower cloud-top $N_d$ for CTHs < 3 km and greater $N_d$ for higher CTHs. The bias above 3 km relative to RSP decreases as turbulent enhancement of collision-coalescence is scaled higher, but below 3 km the low-bias of $N_d$ is persistent no matter the formulation. The increase
in $N_d$ in each of the bin simulations above $\sim 4$ km is due weak precipitation production that accumulates cloud droplets in an outflow layer, driving up $N_d$–an effect that can also be seen in the FIXED_AERO_NO_AC simulation. Median cloud-top $R_{\text{eff}}$ in each of the bin simulations are identical up to 4 km, while above 4 km, BIN_TURB_10X produces larger $R_{\text{eff}}$ as larger precipitation-sized drops are formed. Nonetheless, cloud-top $R_{\text{eff}}$ is consistently low-biased relative to RSP for both the bulk and bin simulations. This apparent bias is explored in the next section using in situ measurements.

**4.4 Comparison of Simulations and In Situ Measurements**

In situ DSDs are evaluated using a methodology that composites instrument measurements of particle size across contiguous horizontal transects with LWC exceeding 0.1 g m$^{-3}$ measured by the Nevzorov hotwire probe, referred to as "cloud passes". Cloud passes were chosen to span a temperature range from cloud base to the 0 °C level with transects long enough to obtain robust sample sizes. A thermodynamic, kinematic, and microphysical summary of the four selected cloud passes is given in
Table 2. The first cloud pass (hereafter CP) was performed in a relatively weak updraft a few hundred meters above cloud base at 19.41 °C with a transect length of $\sim 2$ km. The second CP was a long ascent ($\sim 12$ km horizontally) through a moderate updraft cluster that spanned a temperature range from 15-19 °C (mean temperature of 17.45 °C). The third CP at 7.28 °C was performed $\sim 300$ m below cloud top in a downdraft with a 1-km long transect. Finally, the fourth CP was performed in a relatively strong updraft (maximum $w$ of 9.2 m s$^{-1}$) at 1.04 °C with a transect length of $\sim 1.5$ km. Composite DSDs
are constructed by stitching together the FFSSP, 2D-S10, and HVPS probes following the methodology outlined in Appendix D. For each size bin of a given instrument, the composite DSD is averaged across each 1-Hz sample within the CP before stitching. This methodology yields four continuous size distributions in varying thermodynamic and kinematic environments that are used as an observational target for the simulations. Uncertainties associated with performing composite DSDs are further discussed in Appendix D.

For simulations, we focus only on CNTL and BIN_TURB_10X. Simulated CPs are identified at the temperature level of each observed CP and are selected at output times of 9, 10, 11, and 12 hr. For each cloud pass, the continuous size distribution (including both cloud and rain species for CNTL) is averaged across all grid points within the CP to yield a single sample. Drop size distributions for all cloud passes are shown in Fig. 9 for each temperature level in the leftmost column, along with the observed DSD and the mean of all simulated CPs. The CNTL simulation reasonably produces the two prominent size modes
at $\sim 10$ $\mu$m (so-called "cloud" mode) and $\sim 0.4$-0.5 mm ("rain" mode), with fairly accurate reproduction of the peak $N_d$ in

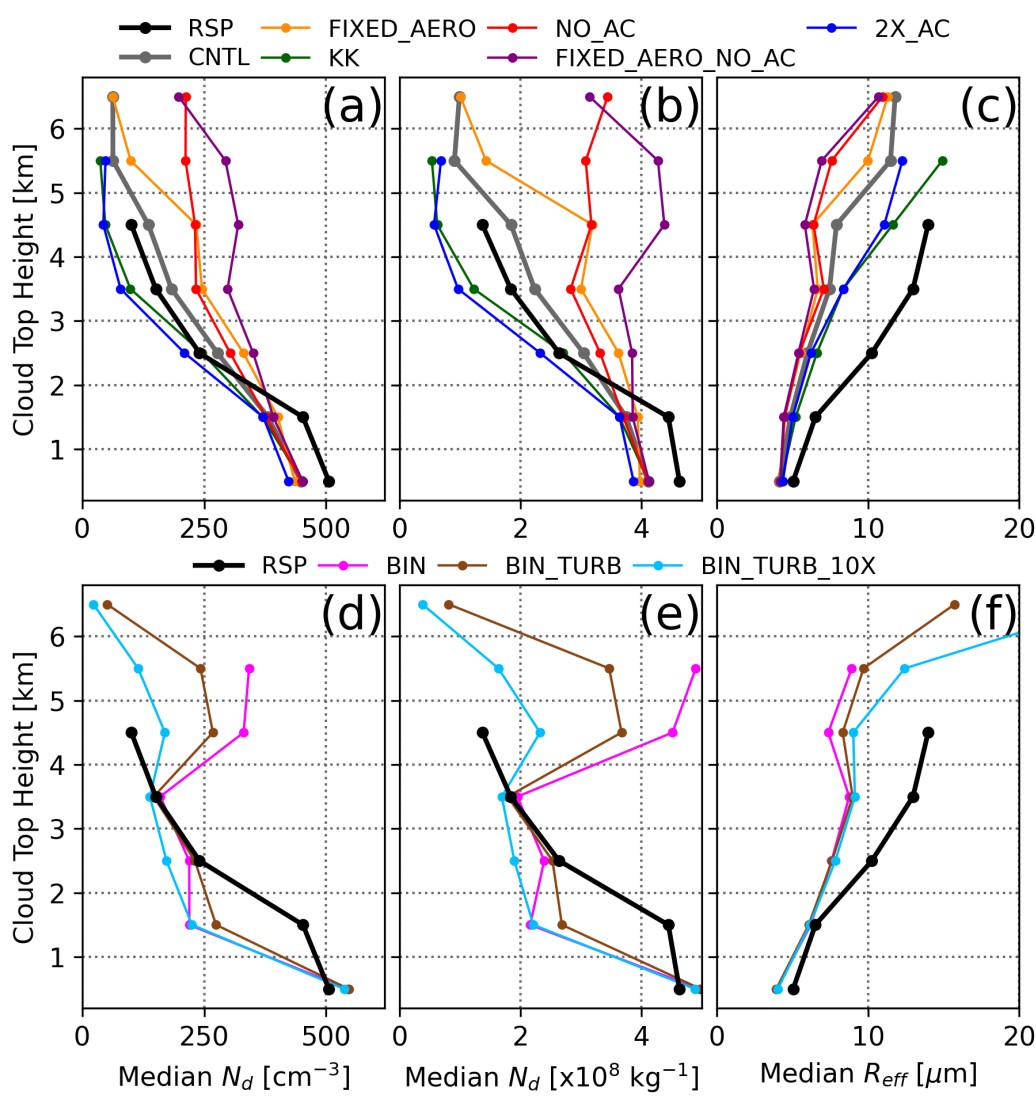

**Figure 8.** Profiles of median (a,d) $N_d$ in units of cm$^{-3}$, (b,e) $N_d$ in units of kg$^{-1}$, and (c,f) $R_{eff}$ as a function of cloud top height for the RSP (black), bulk simulations (top row), and bin simulations (bottom row).







**Figure 9.** Drop size distributions (DSD) from the Learjet (light blue), individual cloud passes from CNTL (black), and the mean DSD from all cloud passes from CNTL (red) for cloud passes identified at a temperature of (a)-(c) 19.41 °C, (d)-(f) 17.45 °C, (g)-(i) 7.28 °C, and (j)-(l) 1.04 °C. The leftmost column shows all cloud passes, the middle column shows only cloud passes with simulated cloud-pass-average vertical velocity ($w$) within 50% of the observed cloud-pass-average $w$, and the rightmost column shows simulated cloud-pass-maximum $w$ within 50% of the observed cloud-pass-maximum $w$. Sample sizes represent the number of simulated cloud passes for a given condition.





**Table 2.** Thermodynamic, kinematic, and microphysical description of four selected cloud passes used to construct composite drop size distributions (DSDs). Cloud passes are defined as contiguous segments with LWC > 0.1 g m$^{-3}$. Values listed for DHARMA are for the CNTL simulation and are from the mean DSD of all cloud passes. Individual DSDs from each instrument and a related discussion is provided in Appendex D.

| CP | Temp. [°C] | | | $w$ [m s$^{-1}$] | | LWC [g m$^{-3}$] | | $R_{\mathrm{eff}}$ [$\mu$m] | | $R_{\mathrm{eff}} < 100\ \mu$m [$\mu$m] | | $N_d$ [cm$^{-3}$] | |
|---|---|---|---|---|---|---|---|---|---|---|---|---|---|
| | Avg. | Min. | Max. | Avg. | Max | Avg. | Max | In Situ | DHARMA | In Situ | DHARMA | In Situ | DHARMA |
| 1 | 19.41 | 19.0 | 19.8 | 0.76 | 1.6 | 0.25 | 0.5 | 8.1 | 10.7 | 7.4 | 10.7 | 348 | 298 |
| 2 | 17.45 | 15.0 | 19.2 | 1.38 | 4.8 | 0.65 | 1.4 | 16.7 | 11.2 | 8.9 | 11.2 | 384 | 277 |
| 3 | 7.28 | 5.9 | 8.9 | -2.82 | -1.0 | 1.15 | 1.8 | 18.5 | 14.4 | 15.2 | 14.4 | 128 | 182 |
| 4 | 1.04 | 0.3 | 2.0 | 4.15 | 9.2 | 0.62 | 0.9 | 55.3 | 15.5 | 19.1 | 15.5 | 52 | 128 |

each mode. The most obvious discrepancy evident in all CNTL CP samples is their more pronounced bimodality and a relative dearth of $N_d$ in the size range between $\sim$ 50-200 $\mu$m (what may be considered a "drizzle" mode), which would be difficult to reproduce in CNTL given the structural separation of species in the bulk scheme. Another notable difference is the slightly narrower distribution of the cloud mode relative to the observed DSDs, as evidenced by nearly all CNTL CPs lying to the left of the observed large-size flank of the cloud mode (rather than spanning it). Nonetheless, the CNTL simulation appropriately captures the observed broadening of the DSD with decreasing temperature (increasing height) demonstrated in Reid et al. (2023).

To further constrain the dynamical conditions of the observed CPs, the middle and rightmost columns of Fig. 9 filter CPs to include only those where the average CP vertical velocity ($w$) is within 50% of the observed average CP $w$ (Fig. 9b,e,h,k) or similarly constrained to be within 50% of the observed maximum CP $w$ ( Fig. 9c,f,i,l). Introducing either dynamical constraint similarly tightens the DSD structure to be closer to the mean DSD, but interestingly does little to alter the primary characteristics of the mean DSD relative to including all CPs. This suggests that the DSD structure here is primarily modulated by microphysical scheme structure regardless of the dynamical conditions.

An identical CP analysis is performed for the BIN_TURB_10X simulation (Fig. 10). The bin scheme also reasonably recreates the structure of the observed DSDs. A persistent difference among all temperature levels is the bin scheme producing a larger number of cloud droplets smaller than $\sim$ 10 $\mu$m, particularly at colder temperatures. This is consistent with the bin simulation's more inefficient precipitation production that results in accumulation of cloud droplets at these levels. As with the bulk scheme, the bin scheme produces a slightly narrower cloud mode. However, the bin scheme produces a pronounced shoulder of the cloud mode at $\sim$ 30 $\mu$m that is most evident at the warmer temperatures. At the coldest temperature (Fig. 10j,k,l), the bin scheme notably produces a more continuous transition from smaller to larger sizes, as in observations, due to the scheme's freedom from parametric constraints. A smaller precipitation mode at all temperatures for sizes greater than $\sim$ 50 $\mu$m is consistent with the bin simulation's relatively weak precipitation production. Similar to CNTL, conditioning the samples based







**Figure 10.** As in Fig. 9 but for the BIN_TURB_10X simulation.



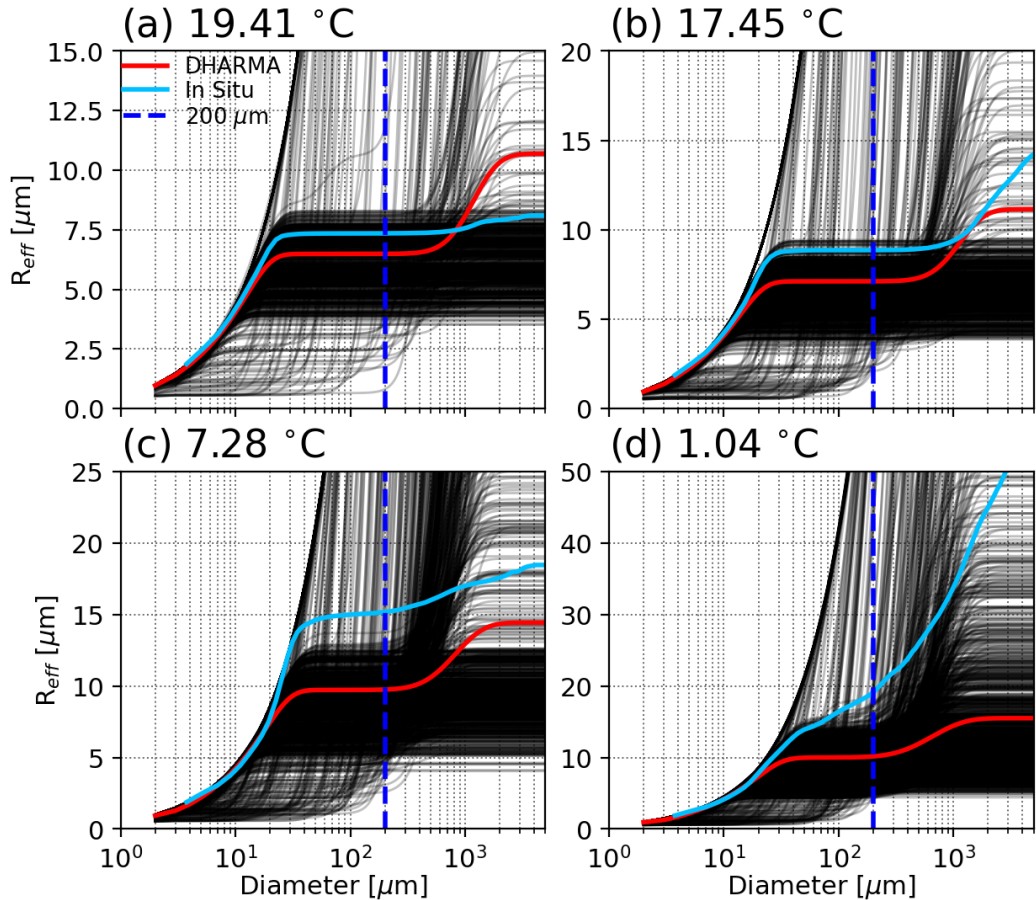

**Figure 11.** Effective radius calculated via cumulative integration of DSDs as a function of diameter for all cloud passes from CNTL (black), the mean DSD (red), and the observed DSD (light blue).

on observed dynamical conditions does little to shift the mean DSD relative to using all simulated CPs, even though it aids in constraining the individual CP DSDs closer to observations (e.g., Fig. 10l).

To determine the potential contribution of these DSD characteristics to the persistent low-bias in cloud-top $R_{\text{eff}}$ presented in Fig. 8, we show the cumulative integrated $R_{\text{eff}}$ for the CP DSDs. For the CNTL simulation (Fig. 11), simulated $R_{\text{eff}}$ is consistently lower than observed $R_{\text{eff}}$ at all temperature levels. Notably, the low-bias in $R_{\text{eff}}$ begins very abruptly at a size of $\sim$ 20-30 $\mu$m, with relatively little contribution by larger sizes until the precipitation mode is reached, and is not particularly sensitive between the small size range and the assumed 200 $\mu$m diameter threshold employed for cloud-top $R_{\text{eff}}$ comparisons

with RSP (dark blue dashed line). Interestingly, this abrupt increase in $R_{\text{eff}}$ contribution occurs well below the obvious dearth in $N_d$ in the drizzle size range (50-200 $\mu$m), implying that the source size region for the bias actually occurs due the slightly narrower distribution of the cloud mode relative to observations.



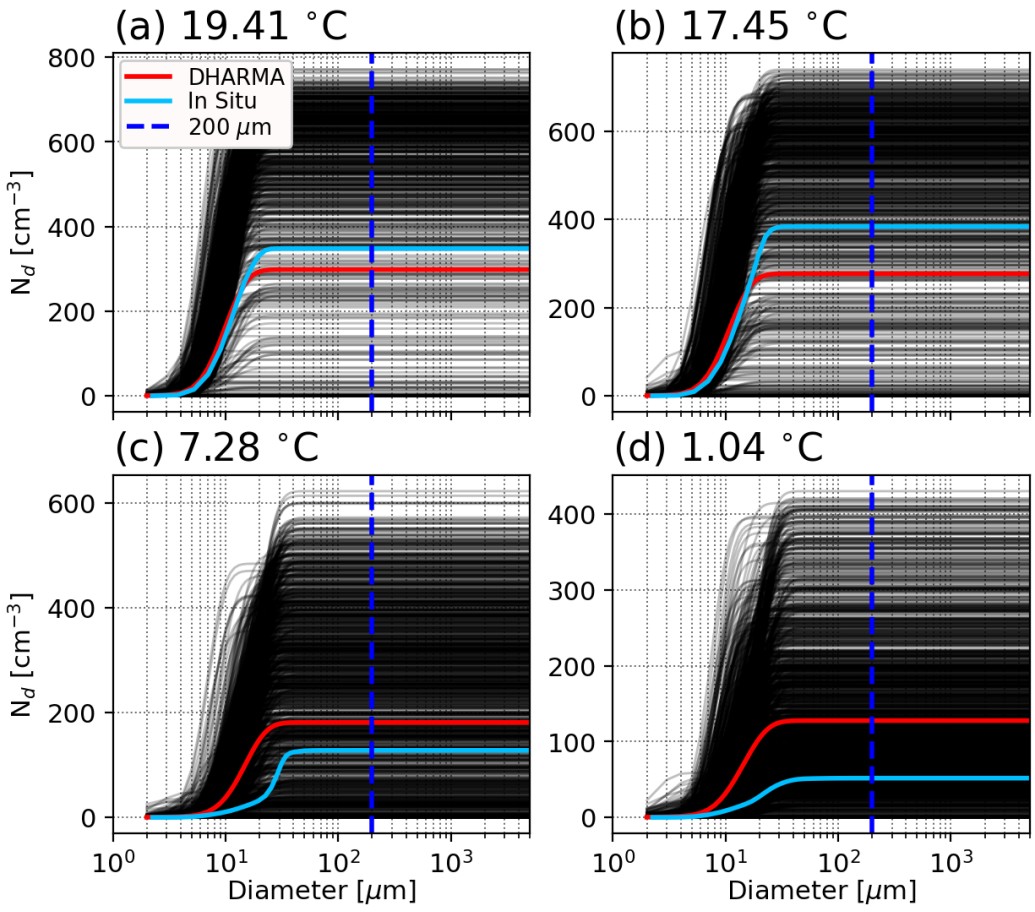

**Figure 12.** As in Fig. 11, but for the cumulative integrated droplet number concentration ($N_d$).

If there is a consistent bias in simulated $R_{eff}$ based on DSD structure, how does this translate to lack of a consistent bias in $N_d$? Similarly to evaluating the cumulative integrated $R_{eff}$, we present the cumulative integrated $N_d$ for CNTL in Fig. 12.

Notably, the cumulative $N_d$ saturates at a size of $\sim 20\ \mu$m at each temperature level with no further contribution by larger sizes. Furthermore, any bias relative to the observed DSDs are not consistent across temperature levels. In fact, Fig. 12 shows that simulated cumulative $N_d$ at warmer temperatures are lower than observed and at colder temperatures are higher than observed, which is consistent with the relative differences between CNTL and RSP shown in Figs. 7 and 8. This suggests that drops within a rather small size range ($\sim 20$-$30\ \mu$m) can drive consistent biases in $R_{eff}$ without the same consistent biases in $N_d$. Note

that the same conclusion was found for the BIN_TURB_10X simulation (not shown).



## 4.5 Thermal-based Evaluation

With a relatively well-constrained bulk microphysics simulation and an understanding of biases rooted in microphysics, we next present a more objective, process-based investigation of DSD evolution at the source of cloud droplet production—cumulus thermals (hereafter referred to simply as thermals; Hernandez-Deckers et al., 2022; Matsui et al., 2024). Thermals are identified

and tracked between hours 9-12 of the simulation for the CNTL and FIXED_AERO_NO_AC simulations only using 1-min simulation output, with the latter simulation chosen to control for the effects of the height-resolved aerosol PSD and collision-coalescence on modulating the $N_d$ profile. The CNTL simulation yielded 392 thermals while the FIXED_AERO_NO_AC simulation yielded 302 thermals. The lower number of thermals identified in FIXED_AERO_NO_AC is likely due to weaker convective dynamics in this simulation relative to CNTL, as discussed below. The mode of thermal lifetime is 3 min, which

is the minimum liftetime to be considered as a valid thermal, with a near-exponential decrease in thermal lifetime at larger values (not shown), which indicates that the majority of detected thermals are rather short-lived and is consistent with results from Hernandez-Deckers and Sherwood (2016, 2018). Thermals are composited in time relative to their maximum ascent rate (defined as time t = 0) and are normalized by their radius (R). The evolution of composite thermals for the CNTL simulation in the X-Z plane between times t = -3 min and t = +3 min (i.e., 3 time steps prior to and after the centered maximum ascent

rate) are shown in Fig. 13. Of all thermals for each simulation, at least $\sim$ 20 % of thermals are represented at times t = -3 min and t = +3 min (or in other words, at least 20 % of thermals last 7 minutes or longer). The evolution of thermals in Fig. 13 are displayed by variables of vertical velocity ($w$), supersaturation (S), cloud water mass mixing ratio ($q_c$), cloud water number mixing ratio ($N_c$; independent of the rain species), and rain water mass mixing ratio ($q_r$).

Thermals in CNTL realize centered $w$ of at least 10 m s$^{-1}$ (Fig. 13a1-a7) and continuous supersaturations at the center of the

thermal throughout their lifetime (Fig. 13b1-b7). The evolution of cloud water mass ($q_c$) shows relatively larger cloud water contents near the center of the thermal lifetime, while near the end (t = +3 min, Fig. 13c7), there is a shearing of maximum regions of $q_c$ towards the right side of the thermal. Consistent with the cloud-top evaluation, $N_c$ decreases during thermal-centered evolution (Fig. 13d1-d7) and $R_{\text{eff}}$ increases (not shown), but $N_c$ also increases slightly at t = +3 min (Fig. 13d7). This feature is also evident in domain-mean in-cloud profiles of $N_c$ in Fig. 6, where $N_c$ commonly realizes local maxima near

cloud-top. One possible explanation for this is that the most undilute thermals reach their thermodynamic level of neutral buoyancy with minimal impacts from entrainment. That is, the $\sim$ 20% of thermals that reach a life cycle stage of +3 min after maximum ascent rate (t = 0) entrain less (and thus experience less $N_c$ dilution) relative to thermals that terminate at a lower altitude. However, this may also be related to the unloading of precipitation from the thermal followed by continued (secondary) activation. Indeed, the evolution of $q_r$ (Fig. 13e1-e7) shows that rain is absent in the initial time-step, reaches a

maximum at t = +2 min with a structure indicating preferred sedimentation on the outer edges of the thermal where $w$ is weaker, and a significant decrease at t = +3 min as precipitation-sized drops exit the thermal. Therefore, persistent supersaturation at the center of the thermal can continue to activate aerosols after rain drops sediment out.

The same thermal evolution is shown for the FIXED_AERO_NO_AC simulation in Fig. 14, but excluding $q_r$ evolution since autoconversion is neglected. Notably, vertical velocities and supersaturations are weaker in this simulation (Fig. 14a1-a7

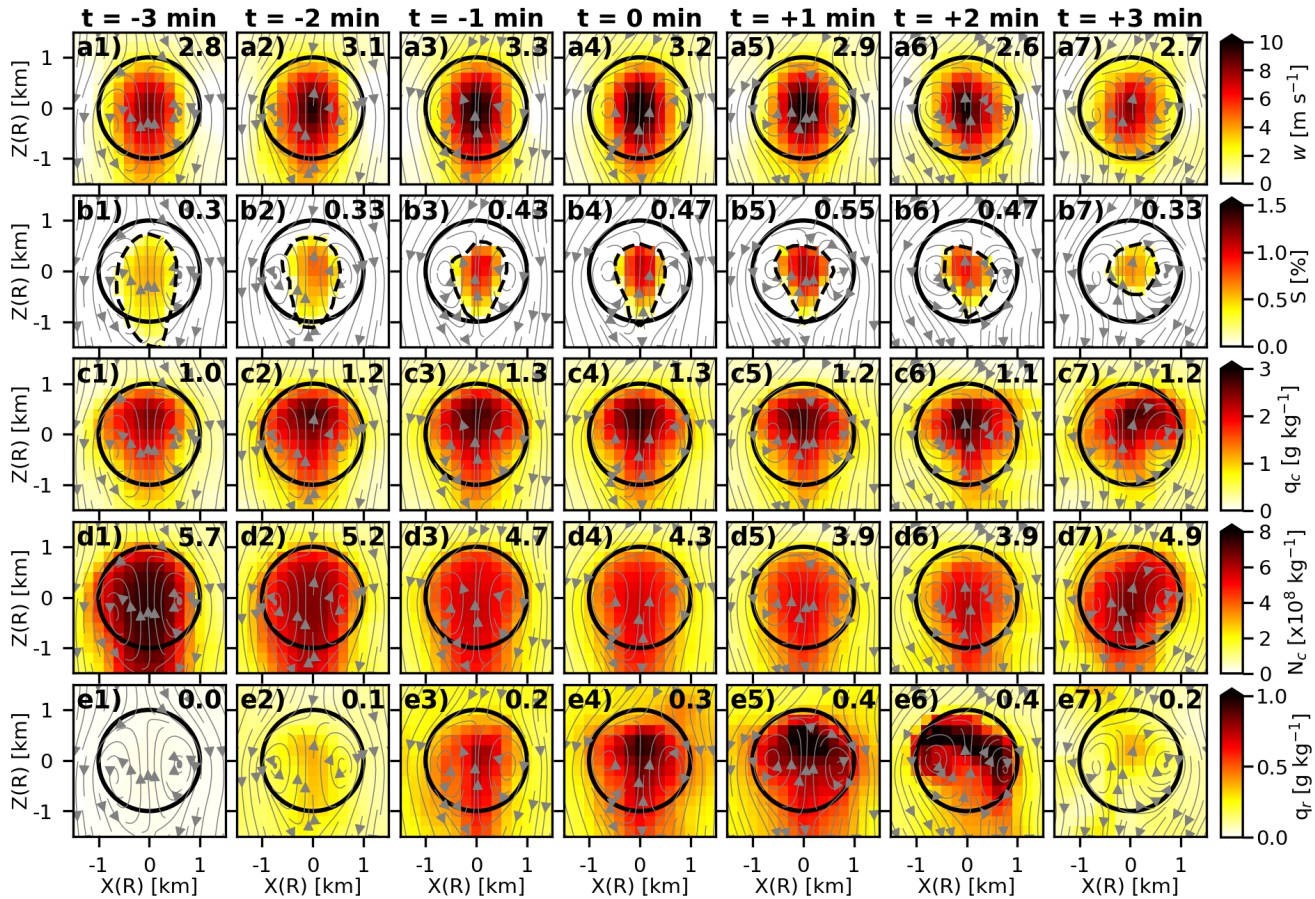

**Figure 13.** Composite thermal evolution (left to right) for the CNTL simulation from 3 minutes prior to maximum ascent rate (t = -3 min, leftmost column) to 3 minutes after maximum ascent rate (t = +3 min, rightmost column), where maximum ascent rate is centered at t = 0 min. Composites are normalized by thermal radius (R) and are displayed in the X-Z plane. Variables are distributed by rows: (a) vertical velocity ($w$), (b) supersaturation (S), (c) cloud water mass mixing ratio ($q_c$), (d) cloud water number concentration mixing ratio ($N_c$), and (e) rain mass mixing ratio ($q_r$). Gray streamlines are perturbation flow and the black ring represents a normalized R = 1. Values in the top right corner of panel are thermal-averaged values for a given time step.



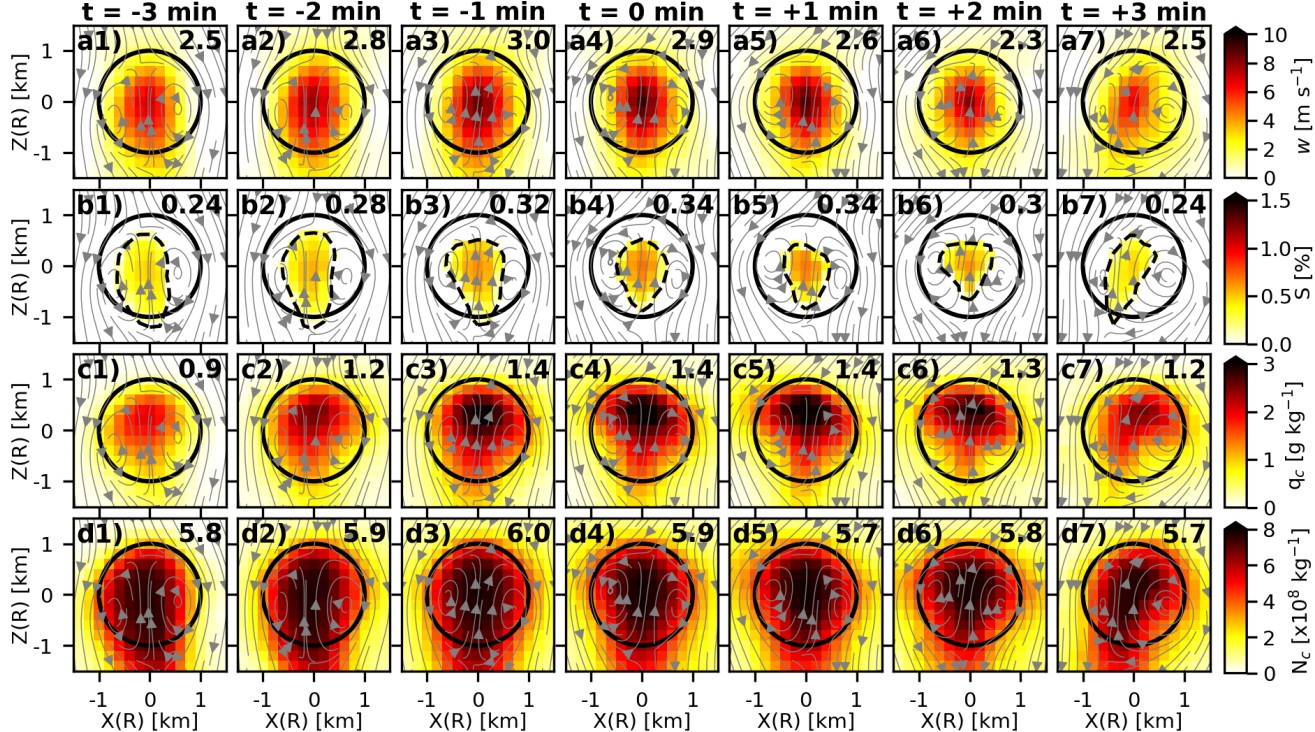

**Figure 14.** As in Fig. 13 but for the FIXED_AERO_NO_AC simulation, with the exclusion of $q_r$.

and b1-bd7, respectively), but supersaturations persist throughout thermal evolution. The $N_c$ evolution (Fig. 13d1-d7) shows a
negligible decrease of $N_c$ throughout the thermal's lifetime, with the clearest difference being the shift in the axis of maximum
$N_c$ due to shear.

These thermal evolutions imply that by controlling for the height-resolved aerosol profile as well as the impact of collision-
coalescence on $N_c$ throughout a thermal's lifetime, the thermal-averaged $N_c$ is largely uniform during ascent. However, en-
trainment is expected to be active in this environment, so why does $N_c$ appear to remain constant? This is investigated via
profiles of averaged thermal characteristics (Fig. 15). The CNTL and FIXED_AERO_NO_AC simulations produce similar
thermal radii between $\sim$ 400-600 m (Fig. 15a). Both simulations also show similar fractional entrainment rates ($\varepsilon$, Fig. 15d;
see Hernandez-Deckers and Sherwood, 2016, for a description of how this is explicitly calculated), which is expected, since
$\varepsilon$ tends to be inversely proportional to radius (Hernandez-Deckers and Sherwood, 2018). However, FIXED_AERO_NO_AC
produces weaker average and maximum $w$ (Fig. 15b,c), which is likely a combination of greater condensate loading as well
as the lack of cold pool development preventing clustering and organization that is realized in CNTL. Profiles of thermal-
averaged $q_c$, $N_c$, and $R_{\text{eff,c}}$ are shown in Fig. 15e-g, along with profile averages of cloudy grid points (LWC > 0.01 g m$^{-3}$)
and convective cloudy grid points (LWC > 0.01 g m$^{-3}$ and $w$ > 1 m s$^{-1}$). No matter the definition, Fig. 15f shows that $N_c$ in
FIXED_AERO_NO_AC is consistently constant with height, similar to the relatively constant cloud-top $N_d$ profile presented



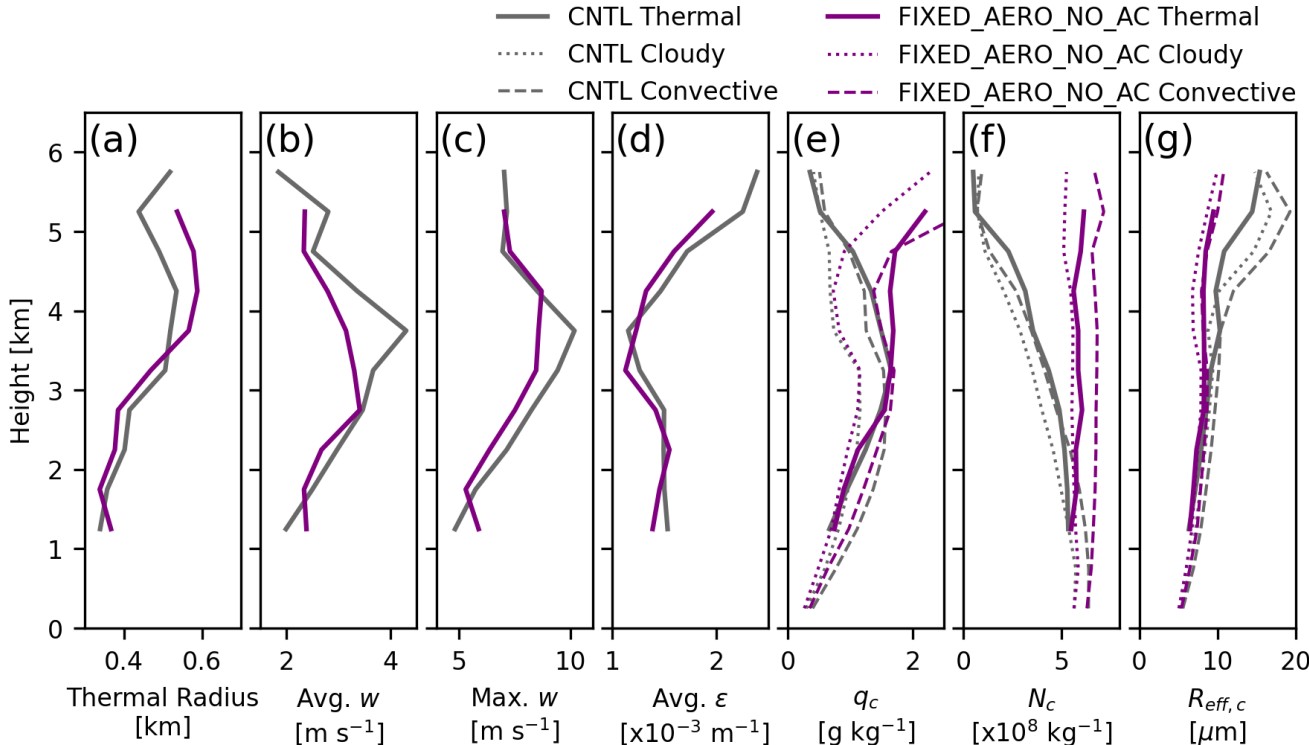

**Figure 15.** Profiles of thermal-averaged (a) radius (R), (b) average vertical velocity ($w$), (c) maximum $w$, (d) average fractional entrainment rate ($\varepsilon$), (e) cloud water mass mixing ratio ($q_c$), (f) cloud water number mixing ratio $N_c$), and (g) cloud water effective radius ($R_{eff,c}$). Profiles are constructed by 0.5-km bins. In (e)-(g), domain-averages for cloudy grid points ($q_c > 0.01$ g kg$^{-1}$) are shown as dotted lines and domain-averages for convective cloudy grid points ($q_c > 0.01$ g kg$^{-1}$ and $w > 1$ m s$^{-1}$) are shown as dashed lines.

in Fig. 8b. This suggests that in the absence of a height-varying aerosol profile and collision-coalescence, the impact of entrainment on reducing $N_c$ is offset by continuous activation of cloud droplets in thermals, for which supersaturations were shown to be present throughout a thermal's entire lifetime in Fig. 14; alternatively, it is possible that secondary droplet activation along cloud edges (i.e., the inflow branch of toroidal circulations) is active herein but was not captured in composites–a process which has been shown to be a relevant aerosol activation mechanism in prior studies (Morrison et al., 2022; Chandrakar et al., 540   2021).



## 5 Discussion

### 5.1 Implications for Developing Convective Microphysics for Large-scale Models

The purpose of any parameterization is to represent a subgrid-scale process as a function of the model's resolved-scale components. While microphysics are ubiquitously parameterized in all numerical models, ESMs must also parameterize convective

dynamics at essentially all scales relevant to cumulus thermals, thus compounding the parameterization problem. While stratiform microphysics are generally more consistent in the current generation of ESMs, following some form of prognostic double moment scheme, current convective microphysics implementations in ESMs vary greatly in sophistication. In the National Aeronautics and Space Administration (NASA) Goddard Institute for Space studies (GISS) ModelE3 (GISS-ModelE3; Cesana et al., 2019, 2021), convective microphysics are diagnostic and the convection scheme uses an entraining two-plume model

that does not distinguish between shallow or deep convection. In this implementation, hydrometeor fall speeds are projected against the vertical velocity calculated from Gregory (2001), and detrained condensate is passed from the convection scheme to the stratiform scheme. The National Center for Atmospheric Research (NCAR) Community Atmosphere Model (CAM) version 5.3 (CAMv5.3) treats the parameterization of stratiform cloud, shallow cumulus, and deep convection differently, with deep convection using the bulk mass flux parameterization of Zhang et al. (1995), which does not explicitly represent cloud

condensate. Instead, it relies on an empirical representation to form precipitation. In one of the more advanced implementations of convective microphysics, a prognostic double-moment bulk microphysics scheme was introduced by Song and Zhang (2011) that includes detailed microphysical processes and the interactions between ice and liquid species. Lin et al. (2023) implemented this parameterization into CAM v.5.3 and, in agreement with Song and Zhang (2011), found an improved representation of stratiform precipitation rates and amount resulting from more realistic convective detrainment in simulations of

tropical deep convection relative to observations. The U.S. Department of Energy (DOE) Energy and Exascale Earth System Model (E3SM) now employs the Predicted Particle Properties (P3) microphysics scheme (Morrison and Milbrandt, 2015; Milbrandt and Morrison, 2016), which enables a continuous representation of ice riming among other departures from Gettelman and Morrison (2015). Wang et al. (2021) found that P3 improves the representation of precipitation from mesoscale convective systems.

There is considerable room for investigating alternative convective microphysics scheme structures, which unlike stratiform microphysics should be closely coupled with convective vertical velocities. Furthermore, recent emphasis has often been targeted towards an improved representation of ice-phase microphysics. Here, we show that in LES of congestus that neglect ice formation, there are deficiencies in the warm-phase component—even with prognostic number concentration for both the cloud and rain hydrometeor species and resolved convective dynamics. In the bulk microphysics scheme presented herein, the width

of the cloud droplet distribution is determined by a fixed dispersion parameter set to 0.3 that directly computes the gamma distribution shape parameter for the cloud droplet species. However, sensitivity tests (not shown) indicated that increasing this dispersion parameter to 0.4, while producing a broader cloud mode, introduced nonlinear effects on the precipitation mode that were unphysical. Therefore, further studies are needed to determine an improved representation of cloud droplet dispersion or perhaps a prognostic method for the cloud droplet shape parameter. Reduced complexity of the aerosol distribution




and mixing state could also be a source of DSD structure and evolution biases. For example, a constant $\kappa$ may be insufficient to represent the size-dependent distribution, for which advanced representations have been suggested (e.g., Su et al., 2010), whereas varying $\kappa$ across different aerosol types (e.g., modes) can improve CCN concentrations relative to observations (e.g., Fierce et al., 2017). Extending the aerosol distribution to larger sizes beyond that employed here (defined by the upper size limit of FIMS—600 nm) and considering the role of giant CCN (e.g. Morrison et al., 2022) may also be important for further

investigating the results herein. Additional modeled case studies that are well-constrained with observations under a diversity of environmental conditions are also needed to confirm how widespread such biases in microphysics schemes may be.

Related to this, it may seem surprising that the bin scheme tested herein produced simulations that diverged further from observations relative to the control bulk microphysics scheme in some leading ways (e.g., $N_d$ profile at the coldest temperatures). It is beyond the scope of this study to determine the precise cause for this. Although the DSD shape has a significantly larger

number of degrees of freedom in the bin scheme, allowing it to better represent the observed continuity between cloud and rain modes in colder CPs, there is a correspondingly greater scope for uncertainties associated with the collision kernel algorithms and efficiencies. However, ultimately the goal of this study is to provide guidance for improved microphysical representation of congestus in ESMs, for which the use of multiple schemes provides a stronger foundation for benchmarking and future interrogation of shared deficiencies. Owing to the computational efficiency of the bulk scheme, it is encouraging to describe

reasonable representation of cloud-top $N_d$ relative to observations. Apparent deficiencies in the $R_{\text{eff}}$ profiles require further study.

## 5.2 Choosing a Benchmark LES Cumulus Congestus Case for ESM Training

Although there are limitations for how the cloud-top $N_d$ is derived from RSP retrievals, future work can be focused on more appropriate derivations for cumulus clouds where assumptions in the adiabatic cloud model break down. Regardless, results

herein are encouraging as we demonstrate a statistically robust constraint for cloud droplet number. With an understanding of established biases in simulated cloud-top $R_{\text{eff}}$, the presented case study can serve as a benchmark LES for simulating the tropical congestus regime. Appropriate thermodynamic forcing, large-scale vertical motion conditions, and aerosol input are provided as supplements that should be used for testing in additional high-resolution models and SCM versions of ESM parent models.

Neggers (2015) showed in an intercomparison of simulated marine stratocumulus-to-cumulus transitions that SCMs can act as a unique fingerprint of their ESM counterparts. For that cloud regime, such a conclusion suggests that the model state, whether it be SCM or ESM, was dominated by the boundary-layer scheme (or "fast physics" relative to the large-scale flow). In a similar manner, the case study presented herein can be used to investigate the interaction of an ESM's convection scheme and coupled convective microphysics. In this framework, simple tests regarding the structural implementation of convective

microphysics may be explored to determine the leading factors of convective microphysics shortcomings in the congestus regime. Furthermore, since this framework proved effective in deriving observed aerosol profiles and harvesting large-scale conditions from mesoscale simulations, additional cases may be constructed from the CAMP²Ex campaign for cases with different aerosol loading and varying convective structures. Indeed, Reid et al. (2023, see their Fig. 6f) show the vast range



of aerosol conditions measured during CAMP$^2$Ex. Ultimately, the CAMP$^2$Ex campaign provided more RSP retrievals than in

any prior campaign and offers a wealth of data that can be used for further evaluation of microphysics in the congestus cloud regime.

## 5.3 Translation to Space-based Platforms and Global Simulations

Of particular relevance to this study is the recently launched Plankton, Aerosol, Cloud, Ocean Ecosystem (PACE) satellite mission (Werdell et al., 2019), which includes two multi-angle polarimeters. The Hyper-angular Rainbow Polarimeter (HARP-2;

Martins et al., 2018) on PACE provides the opportunity to evaluate global distributions of cloud-top microphysical characteristics extending beyond bi-spectral retrievals such as those provided by the Moderate Resolution Imaging Spectroradiometer (MODIS) on the Aqua and Terra satellites. Fu et al. (2022) performed an intercomparison of cloud-top $R_{\mathrm{eff}}$ during CAMP$^2$Ex between MODIS, in situ measurements, and RSP retrievals using both the polarimetric method (as used herein) and its alternative bi-spectral method. They found a persistent high-bias in $R_{\mathrm{eff}}$ using bi-spectral methods by 4-7 $\mu$m in the median and

demonstrated that cloud heterogeneity and 3D radiative transfer effects were at the source of the high-biased bi-spectral $R_{\mathrm{eff}}$ retrieval. This means that the magnitude of the high-bias in bi-spectral retrievals is comparable to the magnitude of the low-bias in simulated cloud-top $R_{\mathrm{eff}}$ relative to RSP presented herein, potentially perpetuating differential interpretation. Such a bias can have important implications for evaluating ESMs that ubiquitously employ bulk microphysics schemes. For example, $R_{\mathrm{eff}}$ is used directly as input to space-based lidar simulators for calculating the particle backscatter coefficient (Chepfer et al., 2008;

Cesana et al., 2021). Moreover, the direct use of $R_{\mathrm{eff}}$ in ESM radiation schemes will propagate error for comparison with satellite retrievals. Furthermore, just as RSP, HARP-2 has the potential of inferring full drop size distributions (without assuming a gamma shape) using the Rainbow Fourier Transform algorithm (Alexandrov et al., 2012b; Reid et al., 2023; Sinclair et al., 2021), which may provide more detailed information similar to that obtained from the in situ DSDs in this study. In addition to polarimetric cloud retrievals, PACE also provides detailed polarimetric aerosol retrievals (Hasekamp et al., 2019; Gao et al.,

2023) which opens the opportunity to study congestus development as a function of aerosol loading and type. We note, however, that the polarimeters on PACE will provide retrievals on a spatial scale of about $5.2 \times 5.2$ km$^2$ which is more than an order of magnitude coarser than the RSP observations used in this study. Understanding of the effect of increased footprint size could be advanced by using forward simulation approaches and model output from case studies such as this that are also well constrained by in situ and RSP measurements.

## 635  6   Conclusions

We present analysis of a tropical cumulus congestus case study simulated using large eddy simulations with observed trimodal aerosol size distribution profiles as input and evaluated against airborne polarimetric retrievals and in situ cloud microphysics measurements. Novel retrievals from the Research Scanning Polarimeter (RSP) provided a statistically robust continuous profile of cloud-top effective radius ($R_{\mathrm{eff}}$) and effective variance ($\nu_{\mathrm{eff}}$), which used together allowed the derivation of cloud-top

drop number concentration ($N_d$). Simulations were performed using both bulk and bin microphysics schemes, and the sensi-



tivity of drop size distribution (DSD) evolution to the collision-coalescence process, the height-resolved aerosol profile, and dilution via entrainment was quantified. A thermal-tracking framework was then used to objectively describe the microphysical evolution at the source of cloud droplet production. To our knowledge, this study provides one of the most highly-constrained evaluations of $N_d$ in cumulus congestus to date. The primary conclusions are summarized as follows:

- Polarimetric retrievals and in situ measurements of $N_d$ and $R_{\text{eff}}$ both indicate a characteristic profile of decreasing $N_d$ and increasing $R_{\text{eff}}$ with increasing cloud top height (CTH).

- A control simulation using bulk warm-phase microphysics reasonably reproduces the trend and magnitude of cloud-top $N_d$ evolution.

- Neglecting collision-coalescence in a sensitivity experiment increases median cloud-top $N_d$ from the control experiment by $\sim 70$ % and 240 % at CTHs of 4.5 and 5.5 km, respectively, indicating a powerful control on the observed profile.

- Neglecting the height-variation of aerosol increases median cloud-top $N_d$ by $\sim 70$ % and 60 % at CTHs of 4.5 and 5.5 km, respectively, indicating the additional importance of thermal entrainment transiting aerosol gradients aloft.

- Neglecting both collision-coalescence and the height-variation of aerosol produces a cloud-top $N_d$ profile that is nearly constant with height, indicating that entrainment alone is not driving the observed trend.

- Comparison with in situ DSDs shows that the control bulk simulation reasonably reproduces the observed cloud and precipitation DSD modes, but exhibits a dearth of $N_d$ over diameters of 50-200 $\mu$m due to structural limitations of the bulk scheme. The bin scheme shows a similar dearth at warmer temperatures, but is able to better reproduce a more continuous transition between the cloud and rain size ranges at the colder temperatures.

- Both the bulk and bin schemes produce a DSD cloud droplet mode that is too narrow, specifically in the diameter size range of $\sim 20$-30 $\mu$m, which results in a ubiquitous low-bias in $R_{\text{eff}}$ across all simulations that is evident both at cloud top and in cloud passes versus in situ measurements.

- A low-bias in $R_{\text{eff}}$ exists despite reasonable reproduction of observed cloud-top $N_d$, which is explained by simulations adequately capturing the DSD cloud droplet mode peak at sizes up to $\sim 10$ $\mu$m, which control $N_d$ but not entirely $R_{\text{eff}}$.

- A thermal-tracking framework demonstrates that neglecting both collision-coalescence and height-variation of aerosol leads to a constant profile of thermal-averaged $N_d$ with increasing height regardless of sampling approach and controlling for dilution via expansion, suggesting that in the absence of these two controlling mechanisms, the influence of entrainment on diluting thermal $N_d$ is offset by continuous droplet activation via sustained supersaturations throughout thermal lifetimes.

The case study presented here offers a promising benchmark for ESM simulations of tropical cumulus congestus, which are lacking relative to other convective modes that have been the focus of informative intercomparison projects (e.g., Siebesma



et al., 2003; Sandu and Stevens, 2011; Neggers, 2015; Neggers et al., 2017; Vogelmann et al., 2015; Endo et al., 2015; Lin et al., 2015). Such a task is aided by supplying large-scale thermodynamics and vertical motion derived from mesoscale simulations of the presented case study as well as trimodal, lognormal aerosol distribution profiles. Use of this case as a benchmark will contribute to informing development of coupled convection and convective microphysics schemes in large-scale models that currently exist in highly variable states of sophistication. This framework also lends itself to future global analysis of warm-phase microphysics evolution via the recently successfully launched space-borne polarimeters on the PACE satellite. Furthermore, this case offers a natural transition to the evaluation of ice-phase microphysics in tropical congestus, which is the focus of future work investigating mixed-phased processes in moderately-supercooled cloud-tops simulated using the case study described herein.

*Code availability.* Python plotting scripts used to make all figures can be found on the NASA Center for Climate Simulation (NCCS) data portal: https://portal.nccs.nasa.gov/datashare/giss-camp2ex/. DHARMA source code is maintained at NASA GISS and can be made available upon request. The NASA-Unified WRF (NU-WRF) is maintained at NASA GSFC, and available for public use upon request (https://nuwrf.gsfc.nasa.gov/).

*Data availability.* The NASA Center for Climate Simulation (NCCS) data portal (https://portal.nccs.nasa.gov/datashare/giss-camp2ex/) archives the following data: 3D output from the CNTL and BIN_TURB_10X simulations (other simulations can be made available upon request), input aerosol profiles and distribution parameters, thermodynamic and large-scale vertical motion forcing files, NU-WRF namelists needed for reproduction of the mesoscale simulation, files containing time-height series of domain-mean variables for all simulations presented herein, and thermal data files. Aircraft data from the Research Scanning Polarimeter (RSP), 3rd Generation Advanced Precipitation Radar (APR-3), in situ cloud probes, and Fast Integrated Mobility Spectrometer (FIMS) are available on the CAMP2Ex data repository (https://www-air.larc.nasa.gov/cgi-bin/ArcView/camp2ex).

**Appendix A: NU-WRF Simulation Setup**

A mesoscale simulation used for harvesting large-scale thermodynamic conditions and vertical motion for LES initialization is performed using the NASA Unified Weather Research and Forecasting (NU-WRF; Peters-Lidard et al., 2015) model with the v3.9.1.1 WRF dynamical core. The NU-WRF simulation was initialized with the National Centers for Environmental Prediction (NCEP) Global Data Assimilation System (GDAS) FNL (Final) operational global analysis and forecast dataset with a spatial resolution of 0.25° x 0.25° and a temporal resolution of 6 h (National Centers for Environmental Prediction National Weather Service NOAA U.S. Department of Commerce, 2015). The simulation was integrated for 21 h from 12 UTC on 24 September 2019 through 9 UTC on 25 September. We use a one-way nested grid with an outer domain size of 2400 x 2400 km and a horizontal grid spacing ($\Delta_h$) of 3 km and an inner domain size of 480 x 480 km with $\Delta_h = 600$ m. The outer and inner domains use a time step of 9 and 1.5 s, respectively, and each has 151 stretched vertical levels. We employ the National



Severe Storms Laboratory (NSSL) microphysics scheme (Ziegler, 1985; Mansell et al., 2010; Mansell and Ziegler, 2013) with 6 double-moment hydrometeor species including cloud water, rain, cloud ice, snow, graupel, and hail. Trimodal, lognormal aerosol profiles are treated as described for the DHARMA simulations with activation following Abdul-Razzak and Ghan (2000) using the minimum supersaturation from Morrison and Grabowski (2008b). Other physics include the 2017 Goddard radiation package (Matsui et al., 2020), the Noah-MP land surface model (LSM; Niu et al., 2011; Yang et al., 2011), the Mellor-Yamada-Nakanishi-Niino (MYNN) 2.5 level turbulent kinetic energy (TKE) planetary boundary layer (PBL) scheme (Nakanishi and Niino, 2009), and the 2D horizontal Smagorinsky scheme for horizontal subgrid-scale diffusion (Skamarock et al., 2008).

## Appendix B: Deriving Large-scale Vertical Motion

Large-scale vertical motion ($w_{LS}$) was evaluated by averaging vertical velocity across various subdomains of the NU-WRF simulation on a 600-m horizontal mesh (domain shown in Fig. 4). While $w_{LS}$ was rather variable across the domain and time, a characteristic profile of ascent below $\sim 5$ km and subsidence above was apparent when averaging over a time period of 6 h that corresponded to flight timing ($\sim 0300$-900 UTC). The magnitude of ascent in the lower troposphere ultimately modulated the acceleration of growing CTH throughout simulation integration, but a maximum of 2 cm s$^{-1}$ is consistent with subdomain averages. The idealized profile shown in Fig. 4a proved sufficient and appropriate for realizing a simulation similar to the observed system evolution, as determined by the timing of CTH growth from the simulation initialization. As stated in the main text, we do not consider this profile to be well-constrained, but rather plausible, and note that derivations of $w_{LS}$ are essential to appropriately represent the dynamical evolution of at least the congestus case studied here. Indeed, Fig. B1 shows the evolution of domain-mean LWC for the CNTL simulation and a simulation with no imposed $w_{LS}$. In the simulation without imposed $w_{LS}$ (Fig. B1b), cloud tops initially grow to $\sim 2$ km and stagnate to a maximum of only $\sim 2.5$ km at the end of the simulation. Therefore, imposed $w_{LS}$ is essential to realize a simulation similar to the observed system.

## Appendix C: Resolution Sensitivity Test

Resolution sensitivity was investigated by performing a simulation with identical physics to the CNTL simulation (which uses a horizontal mesh of 100 m) but with a horizontal mesh of 50 m. Time-height series of domain-maximum $w$, domain-mean LWC, and in-cloud domain-mean cloud-species number concentration ($N_c$) are shown in Fig. C1. Differences between the two resolutions are negligible for each property for the purposes of this study. Cloud-top $R_{eff}$ and $N_d$ were also evaluated relative to RSP (not shown) and the 50-m simulation was found to be indistinguishable from CNTL. This test demonstrates microphysical convergence for the CNTL simulation presented herein.



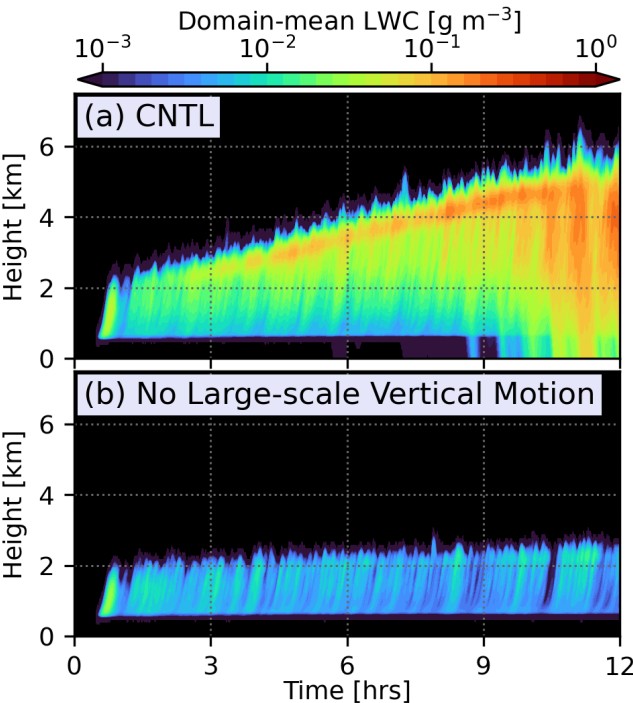

**Figure B1.** Time-height series of domain-mean liquid water content (LWC) for (a) the CNTL simulation and (b) a simulation with the same physics as CNTL but without imposed large-scale vertical motion.

**Appendix D: Composite In Situ DSDs**

Composite observed in situ size distributions are constructed by stitching together instruments in various size ranges. These include the fast forward scattering spectrometer probe (FFSSP; 2-50 $\mu$m with bin width ranging from 1.5 to 4 $\mu$m), the 10 $\mu$m 2D-S (Stereo) optical array spectrometer (2D-S10; 10 $\mu$m - 3 mm with bin width ranging from 10 to 200 $\mu$m), and the high volume precipitation spectrometer (HVPS; 150 $\mu$m - 4.5 mm with bin width ranging from 150 $\mu$m to 3 mm). The size thresholds used to stitch the individual instrument DSDs is chosen based on convergence of overlapping size ranges between

the instruments for a given statistical sample. As shown in Fig. D1, there is considerable uncertainty between overlapping sizes of the smallest size bins for a given instrument and the largest size bins for another instrument. For example, drop numbers in the smallest size bins of the 2D-S10 are 2 orders of magnitude lower than measured by the FFSSP at its largest size bins (Fig. D1a). However, convergence is realized at the upper limit of the FFSSP (50 $\mu$m) such that a continuous distribution emerges. Furthermore, the size threshold is not consistent between different cloud passes and instead depends on the samples

measured by the instrument. For example, the size threshold for stitching the 2D-S10 and HVPS is chosen at 250 $\mu$m for the 19.41 °C cloud pass (Fig. D1a) but at 550 $\mu$m for the 1.04 °C cloud pass (Fig. D1d). Ultimately, the selected cloud passes





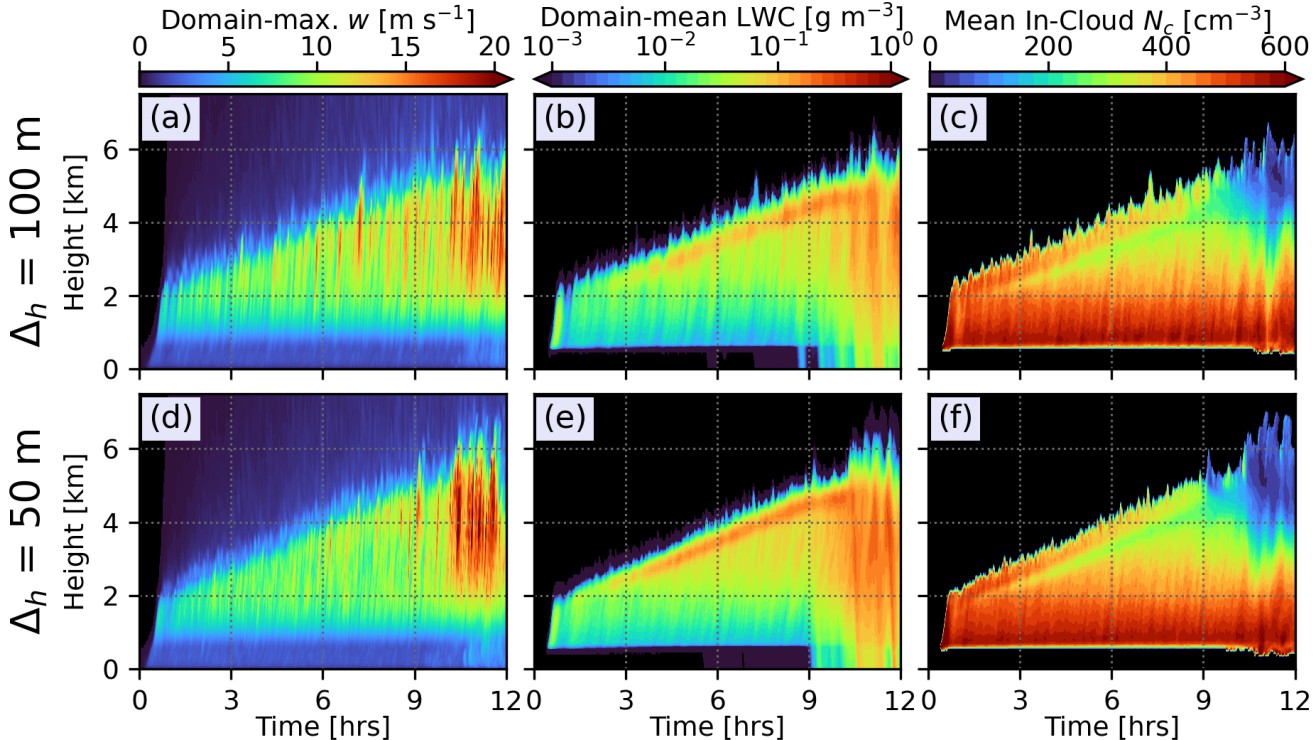

**Figure C1.** Time height series of (a,d) domain-maximum vertical velocity ($w$), (b,e) domain-mean liquid water content (LWC), and (c,f) in-cloud domain-mean cloud-species droplet number concentration ($N_c$) for the CNTL simulation with horizontal mesh of 100 m (top row) and a simulation with a horizontal mesh of 50 m (bottom row).

and size thresholds used to create composite DSDs are chosen to create the most realistic transition between instrument size regions while retaining physical characteristics of the DSD shape.

*Author contributions.* MWS and AMF conceptualized the study. MWS prepared simulations with the help of ASA and AMF (DHARMA) and TM (NU-WRF). Aerosol input data was prepared by QX and JW. Thermal tracking code and guidance was provided by DHD. BvD prepared RSP data and guided interpretation. PL prepared in situ cloud probe data and guided use. MWS performed formal analysis and was primary author of manuscript. All authors contributed to editing the final manuscript.

*Competing interests.* AMF is a member of the editorial board of Atmospheric Chemistry and Physics.





**Figure D1.** Observed drop size distributions (DSDs) for the 4 selected temperature levels discussed in the text. Size distributions from individual instruments are shown as colored lines and the composite DSDs from stitching together instruments are shown as black lines.



*Financial support.* This work was supported by the NASA Atmospheric Composition Campaign Data Analysis and Modeling Program, and
750 the NASA Modeling, Analysis, and Prediction Program.

*Acknowledgements.* Computing support to perform simulations was provided the by NASA Center for Climate Simulations. This work was part of the NASA CAMP2EX campaign (10.5067/Suborbital/CAMP2EX2018/DATA001). We thank Gregory Elsaesser and Kuniaki Inoue for their helpful discussions on convective parameterization, convective microphysics structure, and development pathways. Pacific Northwest National Laboratory is operated by Battelle for the 644 U.S. Department of Energy under Contract DE-AC05-76RLO1830.



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
