# Peer review of "Warm-phase Microphysical Evolution in Large-Eddy Simulations of Tropical Cumulus Congestus: Evaluating Drop Size Distribution Evolution using Polarimetery Retrievals, In Situ Measurements, and a Thermal-Based Framework"

_EGUsphere, 2024_

## Referee Comment (RC2)

Review of "*Warm-phase Microphysical Evolution in Large Eddy Simulations of Tropical Cumulus Congestus: Constraining Drop Size Distribution Evolution using Polarimetery Retrievals and a Thermal-Based Framework*" by Stanford et al. 2024.

The article presents cloud top microphysical properties (droplet effective radius and number concentration) from polarimetry retrievals from the 25 Sep 2019 CAMP2Ex case (cumulus congestus cloud field) and their comparison with in-situ and LES with bulk and bin microphysics schemes. A LES simulation setup was developed based on observed aerosol properties and thermodynamics and dynamics forcing based on NU-WRF mesoscale simulations. The study shows a good agreement between the retrieved droplet number concentration profile and the LES results (with both bulk and bin schemes). However, the effective radius profiles differ quite a bit. The authors attributed it to a narrower cloud droplet distribution mode in simulations compared to observations. They also discussed the importance of droplet collision-coalescence and vertical variability of aerosol distribution in shaping the droplet number concentration profile.

In my opinion, this article presents useful information on the comparison between polarimetry retrievals and LES and the limitations of the bulk and bin schemes in simulating the cumulus congestus cases. The article is written clearly and would be a good contribution to ACP. However, the authors need to address the following points before considering the manuscript for publication:

- "Benchmark" (L2, L593, L669, and several other places): The authors refer to their LES as a "benchmark" LES case. Considering the difficulty in correctly simulating effective radius and overall DSDs, I'm not sure if the current simulations are truly a "benchmark" LES. It would be misleading.

- L80: By "*which implicitly includes nucleation and condensational growth*", are you referring to aerosol nucleation and growth? But that's a transient phenomenon, and you used a constant vertical profile. So, I'm not sure if that's "implicitly" accounted when you used a constant profile. What about aerosol scavenging and processing?

- L94-L100: There's also a precipitating congestus cloud intercomparison case (based on CAMP2Ex) as a part of the International Cloud Modeling Workshop 2024.

- L130-L132: Can the hygroscopicity parameter be separately determined for the three modes, instead of using a mean value? I think it's a big simplification. This assumption might be critical since the droplet number concentration is predicted and compared here.

- L132-L136: That means the $(NH_4)_2 SO_4$ and organics are assumed to be internally mixed. Do you think that's a reasonable assumption based on the observation?

- Fig 1a: Please also add the total concentration profile for reference.

- Eq. 4: Shouldn't an approximate sign not "=" be here since it neglects the skewness of DSDs?

- L191: "*subsequent study*" Missing references?

- L252: Here, the authors mentioned that "*aerosol core mass in liquid drops*" is tracked. How did they do it without using a two-dimensional bin approach? Please explicitly state the assumptions made in solute mass growth through coalescence and reproduction of processed aerosols. The processing and regeneration of large aerosols could be important for a long simulation (12 hours in the current case).

- L270: "*Aerosol activation follows from Ackerman et al. (1995).*" Please explicitly state the assumption with the activation scheme. Since you can't track the hydrated mass growth, do you use the equilibrium assumption (neglects the kinetic limitations)? How did you determine the size of the activated droplet (based on the critical radius of activation or just added to the first cloud droplet bin)? Assumptions related to the treatment of the aerosol activation process and wet growth could be critical for subsequent droplet growth.

- L265: This is the opposite of what's expected due to the numerical broadening discussed in Morrison 2018, Chandrakar 2022, and others. Chandrakar et al. 2022 showed significantly broader DSDs that cause early and more intense rain with a doble-moment bin scheme than a reference run with a Lagrangian microphysics scheme in LES of a cumulus congestus case. Does that mean the current simulations setup or the bin scheme has some issues, compensating errors, or some key elements are missing in the setup?

- L274: Did you use the turbulence enhancement table or directly the theoretical Kernel involving radial distribution function and relative velocity parameterizations (along with collision efficiency enhancement) as a function of Stokes number, settling number, radius ratio, dissipation rate, etc. (including collision efficiency enhancement by turbulence)?

- L321: "*cloud top while in situ measurements are ideally in and near the cloud core.*" - This can't be the reason for the significant difference seen here. Typically, cloud core contains a large number concentration due to less dilution, and cloud top values may not be expected to be that higher (~1.5 times).

- L366-368: Please discuss in detail how the activation is treated in the bin and bulk schemes (maybe in Section 3.2). Please also see my earlier related comments.

- L387: It seems like even for the cases where droplet number concentration is lower (e.g., KK, 2X_AC) the effective radius is lower than RSP and in-situ observations. It potentially indicates some numerical issues, for example, spurious evaporation and secondary activation from numerical diffusion in physical space in Eulerian microphysics schemes (see Chandrakar et al. 2022).

- L426: Maybe "analyzed" instead of "evaluated"?

- Fig 7: Please use a lighter color for simulation median lines (it is hard to distinguish it from RSP). Also, I recommend adding a median line for in-situ data using some finite vertical binning.

- Fig. 8: Please also add a median line for the in-situ data.

- L444-445: To me the cloud mode appeared to be smaller for CNTL compared to obs at and above 17.45 C.

- L446-448: A recent study by Chandrakar et al. 2024 shows a significantly improved representation of the drizzle range embryo drops and an overall great match with in-situ CAMP2Ex observations when a turbulent collision kernel is used in a Lagrangian scheme in LES of one of the CAMP2Ex case. It also significantly accelerates the rain development.

- L453: "*To further constrain the dynamical conditions.*" - Did you also use the same LWC threshold (0.1 g/m3) for obtaining average DSDs from simulations?

- L461-463 and L659-L661: Do the simulations have "a slightly narrow cloud mode" or a shifted mode towards smaller sizes? Can you quantify the difference in spectral width by comparing $D_{std}$? A smaller cloud mode might also be causing a smaller $r_{eff}$ here.

- Fig. 11: Why does the integration of in-situ data start at ~3.5 um? It would be better if you use a consistent minimum size threshold.

- L481-483: Could it be from spurious evaporation and associated secondary activation from numerical issues in Eulerian schemes?

- L655: I do not completely agree with the statement that the cloud mode is captured correctly (also pointed out in my earlier comment). I can see a significant deviation in the cloud mode, especially at 17.45 C and all colder levels above.

**References:**

Morrison, H., Witte, M., Bryan, G. H., Harrington, J. Y., & Lebo, Z. J. (2018). Broadening of modeled cloud droplet spectra using bin microphysics in an Eulerian spatial domain. *Journal of the Atmospheric Sciences*, *75*(11), 4005-4030.

Chandrakar, K. K., Morrison, H., Grabowski, W. W., & Bryan, G. H. (2022). Comparison of Lagrangian superdroplet and Eulerian double-moment spectral microphysics schemes in large-eddy simulations of an isolated cumulus congestus cloud. *Journal of the Atmospheric Sciences*, *79*(7), 1887-1910.

Chandrakar, K. K., Morrison, H., Grabowski, W. W., & Lawson, R. P. (2024). Are turbulence effects on droplet collision–coalescence a key to understanding observed rain formation in clouds?. Proceedings of the National Academy of Sciences, 121(27), e2319664121.

---

## Author Comment (AC1)

**Response to both reviewers**

We greatly appreciate the detailed reviews provided by each of the reviewers, and believe they have improved the quality and clarity of the manuscript. Importantly, we want to first emphasize that during the review process, we found an inconsistency in the presented simulations whereby the aerosol size distribution provided to the model was supposed to use the geometric mean particle radius as input, while we were providing the diameter. This effectively meant that the aerosols were double the size they were supposed to be. We have rerun all of the simulations, and although it did not alter the analysis approach, the results and conclusions changed both qualitatively and quantitatively. In the points that follow (before addressing comments by each individual reviewer), we first list any substantial changes that were made to discussions, figures, and conclusions drawn as a result of these changes to the simulations, along with line/section references corresponding to the *tracked changes* version of the manuscript (not the final revised version). We understand this makes the reviewers' tasks more time-consuming, but hope this manner of documenting the major changes eases the process. In the remainder of the document, we list specific responses to the reviewers in blue text, with line numbers always corresponding to the *tracked changes* version of the manuscript.

**Substantial Manuscript Discussion Changes**

- Most notably, the modifications made to the initial aerosol PSD (which were previously twice the size they were supposed to be) led to significant changes in comparisons between simulated $N_d$ and $R_{eff}$. In particular, $N_d$ for all simulations decreased (less activation now that aerosols are smaller) and larger $R_{eff}$ due to greater precipitation production (ostensibly due to lower $N_d$). These differences propagate to numerous changes in the discussions and conclusions drawn. Substantial changes to the *conclusions* are provided in the sub-heading below. The most significant changes to discussion as a result of the new simulations are in **Sections 4.3** and **4.4**.
- Regarding a comment made by Reviewer #2 for evaluating the DSD spectral width, we realized we were not previously taking full advantage of evaluating the DSD effective variance ($\nu_{eff}$, which is a relative measure of the DSD spectral width), which is also retrieved by the RSP. In general, this gives us greater detail of DSDs that help draw conclusions for the model-observation comparison and explore the ability for the RSP to capture DSDs of different types. We have therefore included additional evaluation of $\nu_{eff}$ where appropriate, which includes additions to Figs. 2, 5, 7, 8, and the new Figs. 9 and 10 (which replaced Figs. 9-12 in the original manuscript, see below for details). We also provide the equation for $\nu_{eff}$ (now Eq. 4) and cloud-top $\nu_{eff}$ (now Eq. 10), which was requested by Reviewer #1. Consequential discussion of $\nu_{eff}$ is scattered throughout the manuscript, with the most significant additions occurring on the following line numbers in the tracked-changes version of the manuscript:
    - **Lines 193-194:** Added analytical expression for $\nu_{eff}$ (Eq. 4).
    - **Lines 381-383, 385-387:** Discuss height distributions of $\nu_{eff}$ from RSP and in situ data.
    - **Lines 470-472:** Added equation for cloud-top $\nu_{eff}$ (Eq. 10).

- **Lines 500-514:** Discuss cloud-top $\nu_{\text{eff}}$ in CNTL and BIN_TURB_10X simulations (Fig. 7).
- **Lines 560-571:** Discuss cloud-top $\nu_{\text{eff}}$ for all simulations, including sensitivity simulations (Fig. 8).
- **Lines 631-641:** Discuss cumulatively integrated $\nu_{\text{eff}}$ for CNTL simulation compared to in situ cloud passes (Fig. 9)
- **Lines 664-684:** Discuss cumulatively integrated $\nu_{\text{eff}}$ for BIN_TURB_10X simulation compared to in situ cloud passes (Fig. 10)

**Figure Changes**

- **Fig. 1:** Added line showing the total aerosol number concentration (sum of all 3 modes) per request of Reviewer #2, and modified line colors in other panels to be consistent with the new line in panel (a).
- **Fig. 2:** Added panel showing time evolution of $\nu_{\text{eff}}$.
- **Fig. 5:** Added panel showing height-dependent distributions of $\nu_{\text{eff}}$.
- **Fig. 6:** Same as original manuscript, but for the new simulations. The maximum value on the colorbar showing in-cloud domain-average $N_c$ time-height profiles is now 400 cm$^{-3}$ and was previously 600 cm$^{-3}$.
- **Fig. 7:** Added panel showing height-dependent distributions of $\nu_{\text{eff}}$ and included new simulations. We also decided to include the simulated "in situ" profiles of $N_d$, $R_{\text{eff}}$, and $\nu_{\text{eff}}$ (i.e., cloudy domain-averages instead of just at cloud-top) in order to compare with the observed in situ values and to demonstrate the difference between simulated "in situ" vs. cloud-top values.
- **Fig. 8:** Added panels showing median profiles of cloud-top $\nu_{\text{eff}}$ for the sensitivity simulations, included the new simulations, and added a line showing the in situ median values per request of Reviewer #2.
- **Figs. 9 & 10:** Replaces previous Figs. 9-12, which showed observed vs. simulated composite DSDs for the bulk scheme (original Fig. 9), the bin scheme (original Fig. 10), and the cumulatively integrated $N_d$ (original Fig. 11) and $R_{\text{eff}}$ (original Fig. 12) at various temperature levels. We decided the discussion and interpretation here made more sense to stack the DSDs alongside the cumulatively integrated $N_d$, $R_{\text{eff}}$, and now $\nu_{\text{eff}}$. This helps to visualize the size-dependence of cumulatively integrated moments juxtaposed on the actual DSD with considerations of size cut-offs relevant for RSP discussion. For this effort, we also now show only cloud passes conditioned on the observed cloud pass average vertical velocity (the middle column in original Figs. 9 & 10) as opposed to showing cloud passes that were not conditioned on vertical velocity at all and that were conditioned on the maximum vertical velocity. We feel that this significantly simplifies the discussion, is an appropriate constraint on the dynamical conditions, and helps to cut down on figures in an already lengthy manuscript.
- **Figs. 11 & 12 (previous Figs. 13 & 14):** Re-ran the thermal tracking analysis on the new simulations.
- **Fig. 13 (previous Fig. 15):** Uses thermal-tracking results from new simulations and adds standard deviations to all profiles, as suggested by Reviewer #1.

- **Fig. 14:** New figure showing average thermal properties as a function of fractional entrainment rate. This was added after attempting to address Reviewer #1's comments regarding discussion of spectral vs. bulk cumulus parameterizations, and to provide evidence of the impact of entrainment on thermal microphysics properties, which was not previously shown.
- **Fig. B1:** Includes new simulations.
- **Fig. C1:** Includes new simulations.

**Substantial Conclusion Changes**

- Differences between RSP cloud-top retrievals and in situ measurements make constraining this case very difficult, especially the feature that cloud-core $N_d$ is lower than cloud-top $N_d$ in observations. We postulate that this is due to breakdowns in RSP retrievals of $N_d$, sparse in situ sampling, and most likely both.
- $N_d$ no longer agrees well with RSP and appears low-biased in all simulations, while $R_{eff}$ in the bin scheme matches very well with RSP and is still a little low-biased in the bulk scheme (though not as much as in the previous simulations). We offer speculation that this is due to both model and observational shortcomings, and provide a detailed discussion of this on **Lines 515-538**.
- We previously had concluded that the simulations produced a cloud mode of the DSD that was too narrow relative to observations that was causing a low-bias in $R_{eff}$. In the new simulations, and with the inclusion of $\nu_{eff}$ analysis, the simulated DSDs appear somewhat too broad with a depressed peak in $N_d$. However we emphasize that this conclusion is only as valid as the reliability of the observational constraints, which appear to be limited in their capabilities for capturing the DSD moments that were a focus of this study.
- Overall, we focus the manuscript more around the ability for the model to bound the observations, discuss limitations for both simulations and observations, demonstrate the capabilities and limitations for using airborne polarimetry retrievals to evaluate LES, and present this as a case study and framework that can be used for future evaluation of congestus in LES and SCMs that are identically forced.
- In accordance with these conclusions changes and our revisions resulting from the reviewers' comments, Section 5 (Discussion) has been mostly rewritten/restructured in general, and the bulleted list in Section 6 (Conclusions) has been significantly modified.

**Reviewer #1**

**Review**

Included here are the more substantial and/or longer comments (i.e., notes, rather than comments, from the attached manuscript file). Grammatical corrections and suggested wording for clarity are included as annotations on the attached manuscript file, but not here. Only the yellow highlighted text and red comments need be addressed directly from the manuscript file. The associated line numbers of those comments are listed at the end of this document.

**General comments:**

The subject of this study is microphysics in tropical cumulus congestus, with a focus on the number concentration and effective radius of cloud droplets near cloud or thermal tops. Retrievals of these quantities at/near cloud top obtained from the airborne Research Scanning Polarimeter (RSP). These were compared to analogous quantities from a set of simulations with bulk 2-moment microphysics and with bin microphysics. In-situ measurements of the DSDs were also compared with simulated DSDs. In addition, a thermal-tracking analysis method was also used to gain insight into the evoluation of microphysical quantities in thermals, and how it varies with microphysics.

The manuscript is generally well written, well organized, and contains excellent graphics. Only minor revisions are recommended.

**The more substantial and/or longer (but still minor) comments:**

**1. line 149:** why is k needed if Reff and veff are both retrieved? Do you want k to relate Rv to Reff?
**Response:** Technically, the derivation for $N_d$ (new Eq. 5) could be represented by replacing the 'k' parameter with the far RHS of (new) Eq. 4 (i.e., in terms of only $\nu_{eff}$). However, the satellite community has historically represented Eq. 5 using the 'k' parameter since it directly relates $R_v$ to $R_{eff}$, and thus LWC and $N_d$ (see derivation in Grosvenor et al. 2018, their Eq. 8). For bi-spectral retrievals, this parameter is a constant, though its natural variability has been documented (e.g., Painemal and Zuidima, 2011). For polarimetric retrievals, however, the relation of 'k' to $\nu_{eff}$ allows the DSD spectral width to vary. Representing the equation in this manner provides consistency with the archetypical bi-spectral retrieval by retaining the 'k' parameter while also highlighting the unique capability of polarimetric retrievals to retrieve spectral width via $\nu_{eff}$. We have added the following discussion on **Lines 199-202** to accommodate this discussion:
"The $k$ parameter has historically been used by the satellite community and is considered constant for bi-spectral retrievals, though the parameter's natural variability has been documented via aircraft observations (e.g. Painemal and Zuidema, 2011). This illustrates a

distinct advantage of the polarimeter's ability to retrieve $\nu_{ef}$ and subsequently implement the *k* parameter into Eq. 5."

**Reference:**

- Grosvenor, D. P., Sourdeval, O., Zuidema, P., Ackerman, A., Alexandrov, M. D., Bennartz, R., et al. (2018). Remote sensing of droplet number concentration in warm clouds: A review of the current state of knowledge and perspectives. Reviews of Geophysics, 56, 409–453. https://doi.org/10.1029/2017RG000593
- Painemal, D., & Zuidema, P. (2011). Assessment of MODIS cloud effective radius and optical thickness retrievals over the Southeast Pacific with VOCALS-REx in situ measurements. Journal of Geophysical Research, 116, D24206. https://doi.org/10.1029/2011JD016155

**2. lines 420-421:** It is not clear how this works; explain. Do you mean to say: weak pcp production allows cloud droplets to accumulate in an outflow layer?
**Response:** Due to the upper-level subsidence (Fig. 3a), insufficient or weak precipitation production caused the clouds formed by rising thermals to accumulate below the subsidence level. Enhancing collision-coalescence via turbulence (and thus enhancing precipitation formation) caused more conversion of that cloud to rain. However, the new round of simulations does not show much of a signal of this feature at all, and thus this discussion has been removed.

**3. lines 455-58.** An alternative conclusion: dynamics 'matters, but its effects are accumulated or averaged. Drops are a product of their histories not their instantaneous environment.
**Response:** Indeed, this is probably a better way to state this, especially because the source of cloud droplet formation here is rising thermals and the DSD shape evolves as a result of that. However, in the revised version of the manuscript, we have since revised the presentation of this discussion by conditioning cloud passes *only* on average vertical velocity. This is both an effort to make the discussion more concise and because we felt that showing the different conditionings was irrelevant to the main points in the study.

**4. lines 516-17:** Fallout of rain would not affect supersaturation, right? But rain is a sink of Nc so increase of Nc due to activation is not clearly evident in presence of rain.
**Response:** That is correct. In the new simulations, this feature of enhanced supersaturation at the end of a thermal lifetime is still present, if not enhanced, and the thermals from the new simulation also precipitate much more heavily (see new Figs. 11-12). However, there is now no clear indication of enhanced $N_c$ at the end of the thermal lifetime, so this speculation is no longer necessary.

**5. section 4.5:** Where in a cloud (relative to the current cloud top) is a thermal typically located? At cloud top?

**Response:** Thermals are not generally going to be at cloud top. One reason for this is that the thermal-tracking algorithm makes a spherical assumption (in order to explicitly calculate entrainment), so the thermal's lifetime ends when it experiences too much deformation, truncating the thermal lifetime prior to the end of actual ascent. Thermals should therefore be thought of as cloud drop sources that lead to the properties observed at cloud top, but should also be considered as distinctly different for most purposes. This is evident especially with the new simulations, whereby Fig. 7 demonstrates significant differences between the cloud-top values and the simulated "in situ" values (or cloud-core value), the latter of which may be considered to be more representative of thermals. We have added the following statement on **Lines 688-691** to more explicitly state that thermals are the regions for the DSD evolution, but should not be considered to necessarily agree with cloud-top values: "While thermal microphysics properties can represent either in-cloud or cloud-top microphysics depending on the locations and lifecycle stages, their successive evolution can provide a source mechanism for droplet activation and the drizzle process, eventually characterizing cloud-top microphysics in convective clouds."

**6.** Fig. 13 might be less cluttered if streamlines were shown in a separate row, and omitted otherwise
**Response:** We feel that the streamlines are best displayed on top of the microphysical variables in order to juxtapose the position of the toroidal circulations with the 2D structure of the microphysical fields.

**7. Figure 15:** Because thermals presumably entrain at different fractional rates, I would expect the variability about the means plotted in Fig. 15 to be large. I recommend giving the reader some idea of the variability among thermals by adding shaded error bars to these plots, for the CNTL and FIXED_AERO_NO_AC profiles.

Ideally, one would stratify the thermal-tracking analysis by fractional entrainment rate. What is plotted in Figure 15 is what a bulk cumulus parameterization would be asked to predict, as opposed to one based on thermals with different fractional rates, such as Arakawa and Schubert (1974), and examined in terms of parcels with different entrainment rates by Lin and Arakawa (1997, Part 2) (https://doi.org/10.1175/1520-0469(1997)054%3C1044:TMEPOS%3E2.0.CO;2).

I recommend making the suggested change to Figure 15 and to add a discussion of what is plotted in terms of a bulk model vs a 'spectral' model.
**Response:** Thank you for providing more information on this, as we hadn't yet considered this perspective. In response, we decided to add a new figure (Fig. 14, discussed on **Lines 749-775**) showing the relationship between thermal properties and the fractional entrainment rate. To first order, this helped to show evidence that entrainment does indeed act to dilute $N_c$ in both CNTL and FIXED_AERO_NO_AC, despite the constant vertical profile of $N_c$ in the profiles (now Fig. 13), which we previously had not explicitly showed. As you state, this figure also aids discussion of the difference between bulk and spectral convection parameterizations, which is now discussed on **Lines 786-809** and also in the Discussion section on **Lines 834-842**.

**8. lines 535-36:** It must be activation of newly entrained CCN. The concentration of CCN from cloud base will be diluted by entrainment, and these CCN alone cannot maintain a constant Nc (per unit mass) with z.
**Response:** See response to next comment.

**9. lines 537-38:** Define secondary activation. If this means activation of newly entrained CCN, then this hypothesis is the same as the first hypothesis. If it means reactivation of CCN from lower levels (i.e., not entrained at the current level), then it cannot offset entrainment dilution.
**Response:** We use secondary activation to encompass any activation of aerosols above sounding-prescribed cloud base. While to first-order this can be considered the activation of newly entrained aerosol, we also mean activation of aerosols that exist in an updraft where acceleration and associated changes in supersaturation can drive subsequent activation. Furthermore, since these congestus clouds are composed of successively rising thermals, activation may occur above the sounding-prescribed cloud base as thermals terminate and new ones form at higher altitudes. However, we do not attempt to discriminate between the different processes through which aerosols activate above the sounding-prescribed cloud base (which would likely only be robustly performed using a Lagrangian scheme). As such, secondary activation is meant to be a somewhat ambiguous definition. We do recognize that our previous description was confusing. We define secondary activation simply as "activation of cloud droplets above cloud base" on **Line 52** (Introduction) and have altered the language on **Lines 743-746** to explicitly define the type of secondary activation that refers to activation of newly entrained aerosols that is relevant to the thermal discussion.

**10. Section 5.1:** This short review of convective microphysics in large scale models may be useful to some readers. However, there are no specific recommendations. It could, and perhaps should, be omitted in the interests of reducing the length of the manuscript.

If you do retain this subsection you may want to point out that adding complexity to convective microphysics schemes usually entails tuning of the microphysical parameters, in particular, rates of conversion to precipitation, rather than using results from cloud resolving models.

Are you aware of any convection microphysics schemes that are aerosol or CDNC aware?
**Response:** We decided to remove this section based on your recommendation, and instead now just briefly discuss convective microphysics development in the Discussion section on **Lines 834-842**, now combining it with the discussion of ESM evaluation in general and a brief discussion of spectral vs. bulk convection parameterizations and how the former can be linked to convective microphysics development. For reference, there are a number of large-scale models employing CDNC-aware convective microphysics, though primarily in research settings as opposed to operational settings. These are provided below.

**References:**
Lin, L., X. Liu, Q. Fu, and Y. Shan, 2023: Climate Impacts of Convective Cloud Microphysics in NCAR CAM5. J. Climate, 36, 3183–3202, https://doi.org/10.1175/JCLI-D-22-0136.1.

Song, X., and G. J. Zhang (2011), Microphysics parameterization for convective clouds in a global climate model: Description and single-column model tests, J. Geophys. Res., 116, D02201, doi:10.1029/2010JD014833.

Storer, R. L., G. J. Zhang, and X. Song, 2015: Effects of Convective Microphysics Parameterization on Large-Scale Cloud Hydrological Cycle and Radiative Budget in Tropical and Midlatitude Convective Regions. J. Climate, 28, 9277–9297, https://doi.org/10.1175/JCLI-D-15-0064.1.

Zhang, J., U. Lohmann, and P. Stier (2005), A microphysical parameterization for convective clouds in the ECHAM5 climate model: Single-column model results evaluated at the Oklahoma Atmospheric Radiation Measurement Program site, J. Geophys. Res., 110, D15S07, doi:10.1029/2004JD005128.

**11. lines 565-566:** Explain/justify/revise this statement.

Condensed water amount is almost entirely a function of altitude not updraft speed.  That is why updraft mass flux is generally sufficient to predict the large-scale convective condensation rate rather than requiring vertical velocity and updraft fractional area separately.

**Response:** Thank you for pointing this out. We have removed this statement and most of this discussion. Just a few statements now remain regarding linking convection parameterizations to convective microphysics, the line numbers for which are listed in response to your Comment #10 and in general in the new Section 5.1.

**12. lines 569-581:**  Before such details as you discuss in this paragraph are included convective microphysics schemes, we may be using convection-permitting models.

**Response:** This is a fair point, but convection-permitting models will still have severe limitations for climate-scale projections on the temporal scale, and not all modeling centers that participate in CMIP/CFMIP/AMIP/etc. projects are actively pursuing convection-permitting climate models. We therefore foresee convective microphysics development for coarse-resolution climate models to be relevant for the foreseeable future. Regardless, this discussion has been significantly revised in general, as stated in the two previous responses, and we have mostly omitted the discussion relevant to this detail.

**13. Lines 582-591:** Perhaps this paragraph should be moved to some other section because the topic seems to differ from that in the rest of the paragraph.

**Response:** This paragraph has been omitted entirely since in the new simulations, the bin and bulk schemes are not consistent in their relative performance relative to observations.

**14. Section 5.2 heading:** "Training" sounds like ML. Is that intended? If not, then perhaps you could use 'Evaluation and improvement' which is of course what training is, but doesn't sound like ML.

**Response:** Changed "Training" to "Evaluation and Improvement".

**Other comments: (see annotations in manuscript file)**

Lines

**Lines that accepted annotated suggestions are followed by a check mark (✅). Those that required further explanation or discussion are provided following the line number.**

**69:** ✅

**80:** We have revised this statement in accordance to Comment #2 raised by Reviewer #2.

**116:** Defined on **Lines 151-152**.

**123:** ✅

**127:** ✅

**Eq. 2:** Added description on **Lines 165-167**.

**145:** ✅

**157:** ✅

**211:** The vertical mean of the large-scale vertical motion mentioned here is over the depth of 10 km, indicating net positive vertical motion. However, this is likely not an important statement for the scope of this study, so we have removed it instead.

**221:** We performed a sensitivity test on a domain size that was doubled (i.e., 38.4 x 38.4 km), as shown here in Fig. R1. A larger domain mostly acted to allow more cold pool development, higher maximum cloud top heights, and larger maximum vertical velocities. However, as shown in Fig. R1, this did not act to significantly alter the microphysical structure of the congestus system, at least in terms of the questions investigated here related to the microphysical vertical profiles. Importantly, a larger domain did not act to enhance cloud-base activation and increase droplet number concentrations. We have added a short statement on **Line 275** and **Lines 529-531** to mention this sensitivity test.

[Figure]

**Figure R1.** Time-height series of (a,e) domain-maximum vertical velocity, (b,f) domain-mean liquid water content, (c,g) domain-mean in-cloud cloud droplet number concentration, and (d,h) domain-mean in-cloud cloud droplet effective radius for the 19.2 x 19.2 km domain (top row) and a 38.4 x 38.4 km domain (bottom row).

**254:** ✅

**282-284:** We have added more clarification on **Lines 348-351**.

**291:** ✅

**343:** ✅

**361-362:** No longer relevant with new simulations.

**379:** Revised, but defined earlier in the manuscript to aid flow/discussion (Eq. 4).

**396:** ✅

**399 (3 comments):** Sentence no longer relevant with new simulations.

**412:** ✅

**419:** No longer relevant with new simulations.

**420:** No longer relevant with new simulations.

**425:** ✅

**429:** Added sample size and leg lengths to Table 2.

**460:** ✅

**497:** ✅

**498:** ✅

**500:** ✅

**504:** ✅

**Figure 13 caption:** Added description of dashed line in caption (new Fig. 11).

**521:** ✅

**560-561:** E3SM uses P3 only for stratiform microphysics, which is now clarified. Convective microphysics are treated much more crudely, so we have removed the mention to E3SM entirely.

**565:** ✅

**568:** We have significantly modified the discussion in this section, so the point here is no longer relevant.

**636 (2):** ✅

**642:** ✅

**643:** ✅

**645 (2):** ✅

**650:** ✅

**653-654:** ✅

**664:** ✅

**667:** ✅

**674:** We have altered this statement (now on **Lines 936-938**) to simply imply improving convective microphysics parameterization rather than "coupled" convection parameterization and convective microphysics.

**Reviewer #2**

**Review of "Warm-phase Microphysical Evolution in Large Eddy Simulations of Tropical Cumulus Congestus: Constraining Drop Size Distribution Evolution using Polarimetry Retrievals and a Thermal-Based Framework" by Stanford et al. 2024.**

The article presents cloud top microphysical properties (droplet effective radius and number concentration) from polarimetry retrievals from the 25 Sep 2019 CAMP2Ex case (cumulus congestus cloud field) and their comparison with in-situ and LES with bulk and bin microphysics schemes. A LES simulation setup was developed based on observed aerosol properties and thermodynamics and dynamics forcing based on NU-WRF mesoscale simulations. The study shows a good agreement between the retrieved droplet number concentration profile and the LES results (with both bulk and bin schemes). However, the effective radius profiles differ quite a bit. The authors attributed it to a narrower cloud droplet distribution mode in simulations compared to observations. They also discussed the importance of droplet collision-coalescence and vertical variability of aerosol distribution in shaping the droplet number concentration profile. In my opinion, this article presents useful information on the comparison between polarimetry retrievals and LES and the limitations of the bulk and bin schemes in simulating the cumulus congestus cases. The article is written clearly and would be a good contribution to ACP. However, the authors need to address the following points before considering the manuscript for publication:

• **"Benchmark" (L2, L593, L669, and several other places):** The authors refer to their LES as a "benchmark" LES case. Considering the difficulty in correctly simulating effective radius and overall DSDs, I'm not sure if the current simulations are truly a "benchmark" LES. It would be misleading.
**Response:** Thank you for pointing out that we do not well define what we mean by benchmark. We have added the following clarification to the manuscript on **Lines 127-131**: "In serving as an LES benchmark case for SCM simulations, we mean to indicate that (1) the meteorological and aerosol set-up is suitable for initializing and forcing the two model types identically, (2) the simulated conditions reproduce basic cloud macroscopic features observed (e.g., cloud top height), and (3) the degree to which simulations statistically reproduce various measurements of cloud microphysical features has been established to the degree possible (e.g., $R_{eff}$)." This is also preceded by discussion of the ICMW case and how the case set-up described here differs. Further mention of this case throughout the manuscript is reworded to specifically refer to its application to SCM simulations.

• **L80:** By "which implicitly includes nucleation and condensational growth", are you referring to aerosol nucleation and growth? But that's a transient phenomenon, and you used a constant vertical profile. So, I'm not sure if that's "implicitly" accounted when you used a constant profile. What about aerosol scavenging and processing?

**Response:** We realize this statement was confusing. We have therefore revised our statement to say the following on **Lines 94 -100**: "Sensitivity tests focus on two processes that exert potentially leading controls on such basic and widely observed profile features: (1) the efficiency of collision-coalescence and its parameterization in different warm-rain formulations and (2) the height-variation of aerosol PSDs. The thermal-tracking framework then examines the role of entrainment and mixing in modulating these profiles." This is immediately followed by a discussion of tests that other studies have performed.

• **L94-L100:** There's also a precipitating congestus cloud intercomparison case (based on CAMP2Ex) as a part of the International Cloud Modeling Workshop 2024.
**Response:** We have added the following brief discussion of this in the Introduction and articulated a key difference between the ICMW case and the one presented here. That is, a case that is suitable for forcing LES and SCMs identically. **Lines 123-127:** "For example, a precipitating cumulus congestus case from the CAMP$^2$Ex campaign was presented for LES and cloud-resolving model intercomparison studies at the 11$^{th}$ International Cloud Modeling Workshop (ICMW) in Seoul, South Korea in 2024. However, a key difference between the ICMW set-up and that presented herein is the former's use of spatially patterned surface heat fluxes as a convective forcing mechanism, which is not straightforward to replicate in typical SCM setups." We again mention the ICMW case as suitable for further exploring structural and numerical limitations, on both the microphysical and dynamical side, on **Lines 525-527**.

• **L130-L132:** Can the hygroscopicity parameter be separately determined for the three modes, instead of using a mean value? I think it's a big simplification. This assumption might be critical since the droplet number concentration is predicted and compared here.
**Response:** We added the following additional discussion on **Lines 169-171**: "Since the AMS does not provide size-resolved chemical composition, $\kappa$ is assumed constant for all three modes. However, ongoing work to derive size-resolved $\kappa$ based on CCN measurements suggests little variability of $\kappa$ with supersaturation (i.e., size)."

• **L132-L136:** That means the (NH4)2 SO4 and organics are assumed to be internally mixed. Do you think that's a reasonable assumption based on the observation?
**Response:** Added the following additional discussion on **Lines 175-180**: "This derivation assumes that aerosols are internally mixed. Characterizing aerosol mixing state was not possible during CAMP$^2$Ex. However, back-trajectories for this flight show exclusively fetches over open ocean, suggesting that the aerosols were minimally influenced by pollution sources (e.g., biomass burning smoke and anthropogenic sources from the metro Manila region). Furthermore, Xu et al. (2021) suggest that the internal mixing assumption in clean marine aerosol environments did not induce any significant error in a CCN closure study." The back trajectories showing exclusive fetches over ocean is provided below, for reference.

[Figure]

**Figure R2.** Altitude-dependent back trajectories from the flight evaluated in this study (RF14).

• **Fig 1a:** Please also add the total concentration profile for reference.
**Response:** Added per request.

• **Eq. 4:** Shouldn't an approximate sign not "=" be here since it neglects the skewness of DSDs?
**Response:** Since this parameter is just a normalized measure of the DSD spectral width without any assumptions about the underlying size distribution, there are no approximations. The effective skewness would be a separate parameter (see Eq. 2.55 of Hansen and Travis, 1974).
**Reference:**
Hansen, J. E. and Travis, L. D.: Light scattering in planetary atmospheres, Space Science Reviews, 16, 527–610, https://doi.org/10.1007/BF00168069/METRICS, 1974.

**L191:** "subsequent study" Missing references?
**Response:** Work is currently underway to extend the simulations here to the ice phase. It is a work in progress, so there is no reference.

• **L252:** Here, the authors mentioned that "aerosol core mass in liquid drops" is tracked. How did they do it without using a two-dimensional bin approach? Please explicitly state the assumptions made in solute mass growth through coalescence and reproduction of processed aerosols. The processing and regeneration of large aerosols could be important for a long simulation (12 hours in the current case).
**Response:** We have added the following description on **Lines 308-310**:

"Core mass is tracked in a droplet size bin by solving a continuity equation for the total dissolved aerosol mass, enabling calculation of the mean solute effect on droplet growth rate in a manner that conserves total solute mass."

• **L270:** "Aerosol activation follows from Ackerman et al. (1995)." Please explicitly state the assumption with the activation scheme. Since you can't track the hydrated mass growth, do you use the equilibrium assumption (neglects the kinetic limitations)? How did you determine the size of the activated droplet (based on the critical radius of activation or just added to the first cloud droplet bin)? Assumptions related to the treatment of the aerosol activation process and wet growth could be critical for subsequent droplet growth.
**Response:** We have added the following description on **Lines 328-331** regarding the bin scheme: "Activation of unactivated aerosol within a bin occurs when supersaturation exceeds the critical supersaturation calculated using the Köhler equilibrium relations. Upon activation, aerosol number is added to the smallest droplet size bin and aerosol mass is transferred to the corresponding droplet core mass bin."

• **L265:** This is the opposite of what's expected due to the numerical broadening discussed in Morrison 2018, Chandrakar 2022, and others. Chandrakar et al. 2022 showed significantly broader DSDs that cause early and more intense rain with a doble-moment bin scheme than a reference run with a Lagrangian microphysics scheme in LES of a cumulus congestus case. Does that mean the current simulations setup or the bin scheme has some issues, compensating errors, or some key elements are missing in the setup?
**Response:** In the new round of simulations, the bin scheme now experiences much more efficient precipitation formation, so this statement is no longer needed.

• **L274:** Did you use the turbulence enhancement table or directly the theoretical Kernel involving radial distribution function and relative velocity parameterizations (along with collision efficiency enhancement) as a function of Stokes number, settling number, radius ratio, dissipation rate, etc. (including collision efficiency enhancement by turbulence)?
**Response:** We directly used the theoretical kernel as detailed in Lee et al. (2021) and similar to the implementation described in Witte et al. (2019). We have modified the following language on **Lines 339-342** to clarify: "In the first experiment (BIN_TURB), the theoretical turbulent collision kernel from Ayala et al. (2008) is incorporated following the implementation described by Lee et al. (2021) that uses the explicitly calculated turbulent kinetic energy dissipation rate ($\varepsilon$) from the subgrid-scale (SGS) diffusion scheme and the collision efficiency enhancement from Wang and Grabowski (2009)".

**References:**
Lee, H., A. M. Fridlind, and A. S. Ackerman, 2021: An Evaluation of Size-Resolved Cloud Microphysics Scheme Numerics for Use with Radar Observations. Part II: Condensation and Evaporation. J. Atmos. Sci., 78, 1629–1645, https://doi.org/10.1175/JAS-D-20-0213.1.

Witte, M. K., P. Y. Chuang, O. Ayala, L. Wang, and G. Feingold, 2019: Comparison of Observed and Simulated Drop Size Distributions from Large-Eddy Simulations with Bin Microphysics. Mon. Wea. Rev., 147, 477–493, https://doi.org/10.1175/MWR-D-18-0242.1.

• **L321:** "cloud top while in situ measurements are ideally in and near the cloud core." - This can't be the reason for the significant difference seen here. Typically, cloud core contains a large number concentration due to less dilution, and cloud top values may not be expected to be that higher (~1.5 times).
**Response:** We agree that the previous discussion did not adequately describe this difference, and we found such a difference rather perplexing relative to expectation and intuition. We mention this difference briefly on **Lines 397-398** in regards to the observational relationship. We then provide a detailed discussion of these confounding differences regarding simulations (which produce the expected relationship) and observational limitations on **Lines 515-538**. This is reiterated on **Lines 888-891,894-895** in the Conclusions. Ultimately, determining the reason for why this seemingly backwards relationship occurs in observations is beyond the scope of our research objectives. Instead, we document it from the available observational data and suggest that such a result requires further evaluation and a better understanding of the observational limitations.

• L**366-368:** Please discuss in detail how the activation is treated in the bin and bulk schemes (maybe in Section 3.2). Please also see my earlier related comments.
**Response:** We have added the following description on **Lines 328-331** regarding the bin scheme:
"Activation of unactivated aerosol within a bin occurs when supersaturation exceeds the critical supersaturation calculated using the Köhler equilibrium relations. Upon activation, aerosol number is added to the smallest droplet size bin and aerosol mass is transferred to the corresponding droplet core mass bin."
and on **Lines 292-296** regarding the bulk scheme:
"Aerosol is activated using Köhler theory following Abdul-Razzak and Ghan (2000) for multiple aerosol modes. This method derives a maximum supersaturation (equal to the critical supersaturation of the smallest activated particles) as a function of dimensionless parameters that include solute effects, curvature effects, and PSD lognormal distribution properties. Here, we use a prognostic supersaturation value after microphysical relaxation that follows from Morrison and Grabowski (2008a)."

• **L387:** It seems like even for the cases where droplet number concentration is lower (e.g., KK, 2X_AC) the effective radius is lower than RSP and in-situ observations. It potentially indicates some numerical issues, for example, spurious evaporation and secondary activation from numerical diffusion in physical space in Eulerian microphysics schemes (see Chandrakar et al. 2022).
**Response:** A general conclusion that we have highlighted more in the revised manuscript is that the available observations are insufficient to robustly constrain these simulations (which resulted in a manuscript title change that uses "Evaluating" instead of "Constraining"). With the

new round of simulations, there are likely issues with both simulated $N_d$ and $R_{eff}$ in the current simulation setup (and realistically with any simulation setup), especially near cloud top, but with the gross disagreements between in situ and RSP measurements and statistically sparse in situ measurements in general, we find it difficult to draw any broad conclusions regarding how and why the model may be underperforming. It is indeed possible that numerical diffusion due to vertical advection in Eulerian schemes and spurious evaporation near cloud top has a significant impact on simulations, among other parameterization and numerics challenges, and we have added this speculation in several places (**Lines 12-13, 518-524, 682-684**, and **894-895**) and discussed the Chandrakar et al. (2022) study (**Lines 521-524**). We stress on **Lines 408-409** that our goal is to bound the observations with our simulations to the extent possible, while not negating that there is also significant measurement uncertainty/undersampling that makes constraint difficult.

• **L426:** Maybe "analyzed" instead of "evaluated"?
**Response:** Changed "evaluated" to "analyzed".

• **Fig 7:** Please use a lighter color for simulation median lines (it is hard to distinguish it from RSP). Also, I recommend adding a median line for in-situ data using some finite vertical binning.
**Response:** Color changed as requested. Because of the addition of simulated "in situ" (or "cloud core") median profiles, we now show only a median line for the observed in situ profiles as well in order to de-clutter the figure.

• **Fig. 8:** Please also add a median line for the in-situ data.
**Response:** Added per request.

• **L444-445:** To me the cloud mode appeared to be smaller for CNTL compared to obs at and above 17.45 C.
**Response:** The DSDs in the new simulations look a bit different, but still each contain their own apparent deficiencies, keeping in mind that bin-wise measurement uncertainties are also not well established. We have altered the language on **Lines 598-600** to more explicitly state that the simulations produce the two prominent size modes to varying degrees of accuracy and suggested possible reasons for this (in both schemes, as detailed in the substantially revised discussion of Section 4.4). Beyond that, we feel the breadth of this study does not have the bandwidth for describing each deficiency with the simulated DSDs in detail since the observations appear to be limited in their capabilities, but instead focus on trying to bound the observations with the simulations.

• **L446-448:** A recent study by Chandrakar et al. 2024 shows a significantly improved representation of the drizzle range embryo drops and an overall great match with in-situ CAMP2Ex observations when a turbulent collision kernel is used in a Lagrangian scheme in LES of one of the CAMP2Ex case. It also significantly accelerates the rain development.
**Response:** Thank you for making us aware of the recent Chandrakar et al. (2024) study. We have added discussion of Chandrakar et al. (2024) in several places to complement the findings

here (**Lines 108-113, 563-570, 649-650,** and **830-832**). However, as shown in Figs. 9-10 of the new manuscript, seemingly small perturbations to the DSD shape can lead to profound differences in $N_d$ and $R_{eff}$. For example, Fig. 3 of Chandrakar et al. (2024) still shows issues with the DSD shape for sizes < 50 um *relative to the observed DSD*. Although adding turbulent enhancement of collision-coalescence in their study helped to enhance the drizzle range and tail of the distribution, we show that those portions of the DSD are not as important for $N_d$ (nor for $R_{eff}$ if the DSDs aren't substantially broad) compared to the contribution from smaller sizes. The differences in peak $N_d$ for sizes < 50 um in Chandarakar et al. (2024, their Fig. 3) may indeed present itself as significant differences in the integrated $N_d$.

• **L453:** "To further constrain the dynamical conditions." - Did you also use the same LWC threshold (0.1 g/m3) for obtaining average DSDs from simulations?
**Response:** Yes, similar LWC thresholds were applied (now stated on **Line 591**). However, to consolidate the discussion of this analysis, we decided to only focus on cloud passes conditioned on their average vertical velocity and LWC as opposed to all cloud passes with no dynamical conditioning.

• **L461-463 and L659-L661:** Do the simulations have "a slightly narrow cloud mode" or a shifted mode towards smaller sizes? Can you quantify the difference in spectral width by comparing D_std? A smaller cloud mode might also be causing a smaller r_eff here.
**Response:** The new round of simulations changed this interpretation entirely. Based on this comment and the ability to also evaluate the effective variance (spectral width) using RSP, we added in evaluation and discussion of this throughout the manuscript (as described in detail at the beginning of this document). The overall message we found by including effective variance is actually that, at least for the cloud mode, the DSDs appear to be too broad (see Fig. 9m-p and 10m-p) relative to observations, with a depressed peak of maximum $N_d$. However, we also consistently emphasize that this is dependent on the reliability of the measurements, which clearly have issues of their own.

• **Fig. 11:** Why does the integration of in-situ data start at ~3.5 um? It would be better if you use a consistent minimum size threshold.
**Response:** This is due to retaining the native grid of the in situ instruments, which is different from the bin scheme, and the bulk scheme's DSDs are purely analytical based on the gamma distributions. Nonetheless, we did check to see if the starting size mattered, and found results to be identical to what is shown here.

• **L481-483:** Could it be from spurious evaporation and associated secondary activation from numerical issues in Eulerian schemes?
**Response:** Yes, this is possible, among other things, particularly insufficient observations. Please see response to earlier comment for our addition of speculation as to why the simulations diverge from observations, with emphasis on both the modeling and observational side.

• **L655:** I do not completely agree with the statement that the cloud mode is captured

correctly (also pointed out in my earlier comment). I can see a significant deviation in the cloud mode, especially at 17.45 C and all colder levels above.

**Response:** We have removed this statement, and instead now point to the differences in simulated $N_d$ and DSDs and offer potential reasons for this. See line numbers in previous comment.

**References:**

Morrison, H., Witte, M., Bryan, G. H., Harrington, J. Y., & Lebo, Z. J. (2018). Broadening of modeled cloud droplet spectra using bin microphysics in an Eulerian spatial domain. Journal of the Atmospheric Sciences, 75(11), 4005-4030.

Chandrakar, K. K., Morrison, H., Grabowski, W. W., & Bryan, G. H. (2022). Comparison of Lagrangian superdroplet and Eulerian double-moment spectral microphysics schemes in large eddy simulations of an isolated cumulus congestus cloud. Journal of the Atmospheric Sciences, 79(7), 1887-1910.

Chandrakar, K. K., Morrison, H., Grabowski, W. W., & Lawson, R. P. (2024). Are turbulence effects on droplet collision–coalescence a key to understanding observed rain formation in clouds?. Proceedings of the National Academy of Sciences, 121(27), e2319664121.

---

## Author Response (AR2)

We greatly appreciate the time and effort taken by both reviewers and the editor to consider and review a substantially revised manuscript relative to the initial submission. Below, we provide a response and associated edits to the one comment raised by Reviewer #2. First, we will address two very minor corrections that emerged from ongoing work on a follow-up study to this one, which have no substantive impact on results or conclusions. All line numbers referenced below refer to line numbers in the *tracked changes* version of the manuscript.

**Minor Edits Unrelated to Review**

1) **Small fix to observed composite DSDs**
   First, we found a very small issue with the observed DSDs that is notable at the smallest size bins for the 2DS10 and HVPS instruments. This issue does *not* pertain to the FFSSP, which was used as the exclusive in situ instrument in Figs. 5, 7, and 8. This issue is most easily shown in a side-by-side comparison of the old and revised Fig. D1 (provided below), which shows the individual DSDs for each instrument as well as the composite used in Figs. 9-10. As shown, a correction to the bin normalization increased the values in the smallest bins (mainly for the 2DS10). The correction provides a better agreement between the instruments in overlapping size ranges. *Importantly, because the instruments were stitched together generally at sizes such that the smallest bins of a given instrument were already neglected in compositing, this makes very little difference in the composite DSDs that were used for analysis in Figs. 9-10*. In the revised manuscript, we therefore include corrected versions of Fig. D1 and Figs. 9-10, but the differences in the latter result in no quantifiable or qualifiable changes needed to the discussion of that analysis. The discussion of Fig. D1 in Appendix D was slightly modified to appropriately discuss the new figure (lines 813-818 in the tracked-changes manuscript).

[Figure]

**Fig. R1.** Original (left) and revised (right) versions of Fig. D1 from the manuscript demonstrating a fix in bin normalization for the 2DS10 and HVPS instruments.

**2) Additional Description of Modified Large-scale Vertical Motion for Bulk vs. Bin Simulations**

We were reminded of a key distinction that needed to be declared regarding the large-scale vertical motion profiles. In previous versions of the manuscript, we neglected to explain that the positive vertical motion below 5 km (shown as a maximum of 2 cm s$^{-1}$ in the original Fig. 3) was adjusted to be a maximum of 3 cm s$^{-1}$ for the bin simulations. This was decided based on numerous sensitivity tests in which the magnitude of the lower troposphere dipole largely acted to modulate the timing of precipitation onset and thus system evolution. As shown in Fig. 6 of the manuscript, the target onset of substantial precipitation production was ~ 6 hrs, which allowed sufficient time for the system to reach peak precipitation production between 9 and 12 hours when the majority of the analysis was performed. For numerous compounding reasons, precipitation onset in the bin scheme was delayed relative to the bulk scheme when using the same large-scale vertical motion profile, and thus a stronger profile was used for the bin simulations to achieve relatively similar precipitation onset with the bulk scheme.

To address this, we have modified Fig. 3 (provided below for convenience) to show the additional profile of large-scale vertical motion below 5 km used for runs with the bin scheme, and have edited the text on Lines 246-249 (tracked changes version) to include this explanation and justification. While it would be possible to include an additional Appendix item describing these sensitivities, we ultimately find them unnecessary for an already lengthy manuscript and based on the justification that constraining large-scale dynamics was not a primary objective of this study, but rather to evaluate microphysics in a congestus system with reasonable evolution.

[Figure]

**Fig. R2.** Modified version of Fig. 3 from the manuscript showing the addition of the large-scale vertical motion line used for bin microphysics runs (dashed line).

**Response to Reviewer #2**

**Overview & Comment:**

**"**The authors present in this paper observations of microphysical properties (droplet size, concentration and DSD shape) in tropical cumulus congestus obtained at cloud top with the Research Scanning Polarimeter (RSP), and in-cloud with the fast forward-scattering spectrometer probe (FFSSP) obtained during the CAMP2Ex experiment. They compare these to a set of large-eddy simulations varying different properties of the microphysical scheme (bulk vs bin microphysics, different bulk schemes, turbulence, and aerosol profile). Thermals were also tracked in the LES to follow how microphysical properties evolve over time.

The paper reads well, the figures are clear and the authors discuss very well the limitations and difficulties of constraining the case they present. I believe the paper to be worthy of publication in ACP.

My only comment would be regarding nu_eff in the simulations: is it actually measuring the broadness of the distribution, or is it measuring its bimodality? Looking at Fig. 9-10, nu_eff has a plateau around 10-100 μm, where it is measuring the broadness of the cloud mode. It rapidly increases near the second mode – the vertical axis is cut off early, but my guess is that it plateaus again after the integral upper bound has passed most of the rain peak, and nu_eff then measures the broadness of the entire DSD. In that regard, I don't think LES values of nu_eff outside of the first or the second plateau are particularly meaningful: they simply measure the presence of a second mode in the DSD relative to an arbitrary size of 100 μm (or 50 μm, as the authors underline). My question would be then: how does this work for the RSP? Since the values reported on Fig. 7 e-f are all below ~0.3, is it much less sensitive to the presence of that peak? I believe some discussion on that would help better understand the apparent discrepancy between measurements and simulations."

**Response:**

We appreciate the reviewer's insightful comments regarding the evaluation of simulated effective variance ($v_{eff}$). To provide further clarity, we include revised versions of Figs. 9 and 10 below (labeled Figs. R3 and R4), in which the cumulatively integrated $v_{eff}$ (bottom row) is displayed on a logarithmic scale. This better reveals the evolution of $v_{eff}$ into the rain mode.

The reviewer was correct in noting that, for the bulk scheme, $v_{eff}$ plateaus to a value representative of the full DSD, particularly once the rain mode is included. In contrast, for the bin scheme—especially under colder conditions where DSDs are broader—$v_{eff}$ increases more gradually with size, without a clear plateau distinguishing cloud and rain modes.

To directly address the reviewer's question of whether $v_{eff}$ reflects DSD broadness or bimodality: the answer is both. Plateau values of $v_{eff}$ can signal the presence of distinct cloud and rain modes, particularly in the bulk scheme and to a lesser extent in the bin scheme at lower

altitudes. However, we emphasize that the plateau value of $\nu_{\text{eff}}$ within the cloud droplet size range remains a useful relative measure of the DSD's breadth. This is most evident in the bin scheme (Fig. R4), where the cloud mode in the simulated DSDs (top row) is clearly broader than in observations. The cumulatively integrated $\nu_{\text{eff}}$ reflects this, increasing more rapidly with size and reaching a higher plateau compared to observations—especially apparent in the previously used linear scale. This interpretation is supported by Fig. 7, where bin-simulated "in situ" profiles show higher $\nu_{\text{eff}}$ at lower altitudes compared to observations, while the bulk scheme agrees more closely—consistent with results in Fig. 9.

In response to the reviewer's suggestion that differences in retrieved and simulated cloud-top $\nu_{\text{eff}}$ may result from the simulated "retrieval" method sampling more of the drizzle/rain mode, we previously noted (lines 464–468 of the manuscript):

"Overall, the differences between simulated cloud-top and "in situ" $\nu_{\text{eff}}$ imply that cloud-top identification using Eq. 10 is rather sensitive to drops in the precipitation size range, even at and near cloud top and to a greater degree at the highest altitudes where both $N_d$ and $R_{\text{eff}}$ appear reasonably well reproduced. Indeed, decreasing the size cut-off for cloud-top distributions in Eqs. 8-10 from r = 100 µm to 50 µm significantly decreased the cloud-top $\nu_{\text{eff}}$ values for both CNTL and BIN_TURB_10X, but did not have a large impact on cloud-top $N_d$ and $R_{\text{eff}}$ (not shown)."

That said, we acknowledge that further speculation is warranted. In particular, we identify three plausible contributing factors, though they are not easily resolved within our current framework:

1. **Instrument sensitivity limitations**: The truncation size threshold used to mimic the RSP is necessarily idealized (e.g., fixed at 100 µm), whereas in reality, the RSP's size sensitivity depends on the extinction properties of the sampled cloud scene.

2. **Cloud-edge resolution in LES**: The model may inadequately resolve sharp cloud-top gradients, allowing drizzle-sized particles to exist near cloud top more frequently than observed.

3. **Homogeneous mixing assumption**: The simulations assume homogeneous mixing during entrainment, preserving droplet number and allowing larger, slowly evaporating droplets to persist near cloud top. This may contribute to artificially broad DSDs within the size range visible to the RSP.

The first two points were already discussed at various points in the manuscript (e.g., lines 466-468, 470-472, 738-740), but we have added the following to lines 477-481 of the revised manuscript as an extension of the paragraph that already discusses potential reasons for these discrepancies:

"Another plausible explanation for the substantially larger cloud-top $\nu_{\text{eff}}$ in simulations at higher altitudes may be related to the model's enforced assumption of homogeneous mixing at subgrid scales, whereby droplet number is preserved

during entrainment. This may allow a broader range of relevant droplet sizes —
including large, slowly evaporating ones — to persist near cloud top, leading to
artificially broad DSDs within the RSP's sensitive size range."

Lastly, we have replaced Figs. 9 and 10 with the updated versions (Figs. R3 and R4 below)
using logarithmic scaling for cumulatively integrated $\nu_{\text{eff}}$ to better illustrate these points. To
accommodate this slight modification and clear up some of the discussion, the discussion on
lines 564-575 and 587-594 has also been modified.

[Figure]

**Fig. R3.** Modified version of Fig. 9 from the manuscript that sets the y-axis of effective variance (bottom row) to
logarithmic scaling.

[Figure]

**Fig. R4.** Modified version of Fig. 10 from the manuscript that sets the y-axis of effective variance (bottom row) to logarithmic scaling.